# Meta-Learning and Meta-Reinforcement Learning - Tracing the Path towards Deep Mind's Adaptive Agent

## Abstract

Humans are highly effective at utilizing prior knowledge to adapt to novel tasks, a capability standard machine learning models struggle to replicate due to their reliance on task-specific training. Meta-learning overcomes this limitation by allowing models to acquire transferable knowledge from various tasks, enabling rapid adaptation to new challenges with minimal data. This survey provides a rigorous, task-based formalization of meta-learning and meta-reinforcement learning and uses that paradigm to chronicle the landmark algorithms that paved the way to DeepMind's Adaptive Agent, consolidating the essential concepts needed to understand the Adaptive Agent and other generalist approaches. It discusses the relevance of meta-learning and meta-RL in the era of scaling foundation models and generalist agents and outlines open problems and future directions.

## 1 Introduction

Humans excel at reusing prior knowledge to adapt rapidly to novel tasks. In contrast, standard machine learning models are typically trained for a single task on a large amount of task-specific data. Optimized for task-specific peak performance, they excel in trained domains but struggle to generalize to new tasks. Meta-learning (or learn how to learn) seeks to extract higher-level knowledge and strategies from a distribution of distinct yet related tasks, enabling models to adapt quickly to new challenges with minimal additional data.

Historically, the conceptual groundwork for self-adapting artificial intelligence was laid by Schmidhuber (1987). Numerous studies in the subsequent decades established the theoretical foundations of meta-learning Sutton (2022), but the field experienced a resurgence, concurrent with the rise of deep learning, when substantially larger datasets and computational resources became available. By introducing Model-Agnostic Meta-Learning (MAML), Finn et al. (2017a) introduced the first landmark algorithm in the class of gradient-based meta-learners, the abstraction of gradient-based learning to the meta-level (see Section 3.1). Around the same time, Duan et al. (2016) introduced $RL^2$, the first landmark of the class of memory-based learners (see Section 3.2). Since then, meta-learning techniques have diversified across few-shot image classification He et al. (2023), Gharoun et al. (2024), Li et al. (2021), neural architecture search Elsken et al. (2019), Hospedales et al. (2022), Ren et al. (2021), Pereira (2024), neural language processing Yin (2020), Lee et al. (2022), Lee et al. (2021), reinforcement learning (see Section 2.2), and application fields like

- robotics, where meta-learners learn novel strategies from only a few demonstrations Finn et al. (2017b), or experiences Johannsmeier et al. (2019).

- health care Rafiei et al. (2024), where patient- or disease-specific data is often sparse Tan et al. (2022), Maicas et al. (2018).

- adaptive control McClement et al. (2022), Duanyai et al. (2024), where system parameterizations change over time.

Even for the preparation of space missions Gaudet et al. (2020), researchers use meta-learning to pre-train models that can rapidly adapt in real time to changing environmental conditions. Consequentially, several

works survey meta-learning by categorizing different sub-classes of meta-learning Beck et al. (2023b), Vettoruzzo et al. (2024), differentiating between the numerous related topics Vettoruzzo et al. (2024), Barcina-Blanco et al. (2024), Upadhyay (2023), addressing open problems Beck et al. (2023b), or reviewing overlaps between meta-learning and its most related fields Upadhyay (2023).

However, the literature lacks a compact, rigorous mathematical treatment tying meta-learning theory to practical implementations. Performance measures are frequently illustrated informally (e.g., in figures or captions) but are rarely defined mathematically, which complicates fair comparisons between methods. Additionally, practical concepts such as validation or meta-validation are seldom formalized. This issue is particularly acute in meta-RL, where agents' actions influence data collection; accordingly, cumulative reward must be explicitly incorporated into the meta-objective. Yet existing surveys on meta-RL (e.g., Beck et al. (2023b)) do not provide a clean mathematical derivation of meta-RL from supervised meta-learning.

But a careful understanding of meta-learning and meta-RL paradigms is highly relevant - perhaps more than ever. As large black-box foundation models and generalist agents scale, they increasingly exhibit emergent capabilities that reproduce behaviors previously engineered via bespoke training schemes (see Section 4). Without precise formalisms and metrics, it remains difficult to assess whether and to what extent such capabilities constitute genuine meta-learning.

**Contribution and Paper outline**

The primary goal of this work is to provide a rigorous formalization and historical synthesis of meta-learning and meta-RL within the task-based paradigm introduced in Section 2. Although meta-learning is distinguished from its closest neighbors in appendix A, fields like continual learning, self-supervised learning, and active learning remain untouched. The curious reader is therefore referred to Vettoruzzo et al. (2024) or specific surveys instead. In contrast, the central contribution of this survey is a chronological presentation of the landmark developments culminating in DeepMind's Adaptive Agent (ADA), using the paradigm of Section 2 as a unifying framework to consolidate the essential knowledge required to understand ADA and related generalist approaches. For this purpose, this work distributes as follows:

- We derive meta-learning from standard supervised learning (Section 2.1) and systematically transfer the resulting notions and formulas to the meta-RL setting (Section 2.2). As part of this derivation, we introduce precise performance measures to provide a unified mathematical framework for concepts that are often treated informally in the literature.

- In Section 3, we apply the formal paradigm as a consistent interpretive lens to present the landmarks on the path from early meta-learners to ADA. The paradigm of Section 2 serves as the organizing thread: each new algorithm is situated within the same formalism to clarify how the field evolved. We begin with the simpler family of gradient-based meta-learners (Section 3.1), then present memory-based (black-box) algorithms. The memory-based sequence follows a timeline from simpler to more complex architectures: RNN-based meta-learners (Section 3.2), transformer-based meta-RL (Section 3.3), and finally the Adaptive Agent (Section 3.4), which integrates concepts from the preceding subsections.

- We analyze ADA as a representative meta-RL instance of a large-scale generalist agent. We discuss the key scaling and enhancement techniques (e.g., distillation, automated curriculum learning) and position these mechanisms within the formal paradigm to explain how they contribute to ADA's capabilities.

Finally, we examine emergent capabilities, interpretability challenges, and open problems in Section 4, before giving an outlook framed by the "three pillars of general intelligence" and concluding our work in Section 5.

## 2 Paradigm

This section formally introduces meta-learning and meta-RL by comparing them to standard machine and reinforcement learning respectively. It, thereby, creates a consistent structure of terms and notions used throughout the rest of the entire work.

### 2.1 Meta-Learning

In standard machine learning a learner $f_\theta$ with parameters $\theta$ is trained to solve a particular task $T$ of the form [1]

$$T := (\mathcal{L}, \mathcal{X}, \mu, \mathbb{T}, h), \tag{1}$$

by minimizing the loss function $\mathcal{L} : \mathcal{X} \to \mathbb{R}$ on some training data $X_{\text{train}}$ out of the observation space $\mathcal{X} \subseteq \mathbb{R}^d$. As this work focuses on deep learning, the parameters $\theta$ always correspond to the weights of a deep neural network.

**Example 1 - Image Classification:**

Let $X$ be a labeled image set out of the observation space $\mathcal{X}$ of images showing cats or dogs, and let $X_{\text{train}}, X_{\text{val}}$ and $X_{\text{test}}$ be disjoint subsets of $X$ with a split of 80% training data and 10% validation and test data. Then, the task $T$ to classify images between 0 (dog) or 1 (cat) is formally defined by defining the components in (1), i.e.,:

- The loss function $\mathcal{L}$ is the cross entropy loss. However, any suitable loss function can be selected.

- The initial distribution $\mu$ is the unit distribution over all images in $X_{\text{train}}$ as all images are equally likely to be selected first.

- The transition $\mathcal{T} : \mathcal{X} \times \mathcal{X} \to \mathbb{R}$ represents the transition from one observation to another. Here, it is a unit distribution conditioned on all images selected so far since all remaining images in $X_{\text{train}}$ are always equally likely to be chosen next.

- Since each shot in image classification consists of only one image, the horizon $h$ equals 1. [a]

The goal is to find the optimal function $f^*$ perfectly classifying any image as cat or dog respectively [b] by minimizing $\mathcal{L}_\theta$ w.r.t. $\theta$. The notation $\mathcal{L}_\theta$ denotes that observations $x \in \mathcal{X}$ are processed or collected via the function $f_\theta$. The learner $f_\theta$ can be any suitable machine learning model, e.g., a convolutional neural network processing images to output whether they contain a cat or a dog. The validation and test sets $X_{\text{val}}$ and $X_{\text{test}}$ contain labeled cat and dog images excluded from training. They allow to evaluate whether the learner $f_\theta$ truly learned to distinguish cats and dogs or only overfitted by memorizing the images in $X_{\text{train}}$. Validation runs during training to monitor progress; testing runs once after training to report final performance. But this way of testing the model $f_\theta$ does not provide any information if the same learner could also distinguish between cats and horses since the latter class does not exist in the task's observation space $\mathcal{X}$.

---

[a]Note that horizon, transition, and initial distribution play a stronger role in the reinforcement learning setting presented in Section 2.2.

[b]The optimal function $f^*_T$ is not necessarily unique, but can, without loss of generality, be assumed as one particular function, as each optimal function, is per definition, as good as another one.

---

The loss function $\mathcal{L}$ as well as all other task-specific components like observation space $\mathcal{X}$ or transition $\mathbb{T}$ remain unchanged throughout the whole training and testing process. As a consequence, the model $f_\theta$ is tailored to particularly solve the task $T$, while it generally fails to generalize to out-of-domain tasks, i.e., to tasks with different loss, transition, or observation space Upadhyay (2023). However, for many tasks the number of training examples available is not sufficient for learning, and sometimes training a (standard)

---

[1]This formalization of a task is inspired by Finn et al. (2017a), Upadhyay (2023), Rakelly et al. (2019).

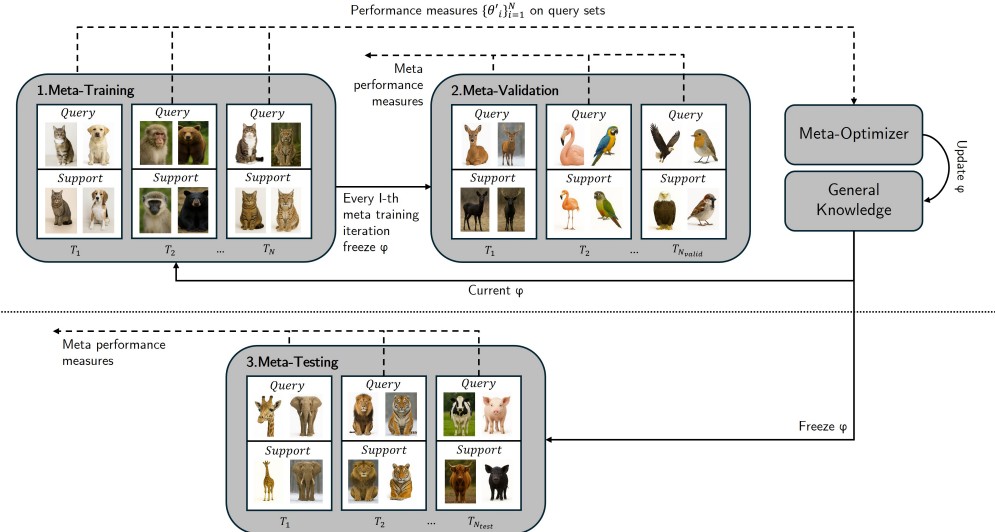

Figure 1: Meta-learning of 2-way 1-shot animal classification tasks. The current meta-knowledge $\varphi$ is the prior for one-shot learning of each particular classification task. During meta-training, the meta-optimizer receives all $N$ query set losses of the adapted models to update meta-knowledge $\varphi$. Meta-validation evaluates the training progress on new classification problems every $l$ meta-epochs, while meta-testing on unseen classifications takes place after meta-training.

learner for a new but very similar task fully from scratch is too costly in terms of computation power or training time.

For these reasons, meta-learning - in contrast to standard machine learning - aims to collect general knowledge over a family of similar tasks, aiming for the best possible fine-tuning for each task within only a few examples, i.e., adapting to a task within $K$ shots rather than with a full training [2]. Modifying the observation space $\mathcal{X}$ of Example 1 to contain labeled images of several different animals, but only ten images per class, one obtains a meta-learning problem. With the same training-validation-test split as in Example 1, this means eight images per class for training and only one for validation and testing, respectively. This small amount of training data does not suffice for training a standard learner properly. However, on a meta-level, distinguishing between all these different animals does not differ much from the task in Example 1. Quite the opposite is the case since, on an abstract level, classifying dogs vs. cats requires similar high-level skills as classifying cats vs. horses: The model needs to identify the creature in the picture first, recognize shapes of noses, ears, or other body parts, examine the silhouette, etc. In other words, classifying cats vs. horses is just another task from the task distribution over different animal classification tasks of the form presented in Example 1. In this way, the meta-task to classify any animal can be separated into several simplified tasks that share some common, high-level structure and belong to the same "meta-problem" of classifying animals. The following paragraphs formalize the meta-learning paradigm, while Figure 1 applies it to the meta-problem of classifying different animals against each other.

**The Meta-Learning Paradigm**

Formally, the meta-learning paradigm Thrun & Pratt (1998) consists of a distribution $p(T)$ over tasks of the form

$$T_i := (\mathcal{L}_i, \mathcal{X}, \mu_i, \mathbb{T}_i, h). \tag{2}$$

Each task of the form (2) (possibly) has its own loss function, transition dynamics, or initial distribution, while the observation space $\mathcal{X}$ and the horizon $h$ are generally assumed to be equal between tasks from the same distribution $p$ (see e.g., Finn et al. (2017a), Rakelly et al. (2019)). This work follows that convention,

---

[2]If only one shot or even no shot is used for fine-tuning, one speaks of one-shot or zero-shot learning, respectively.

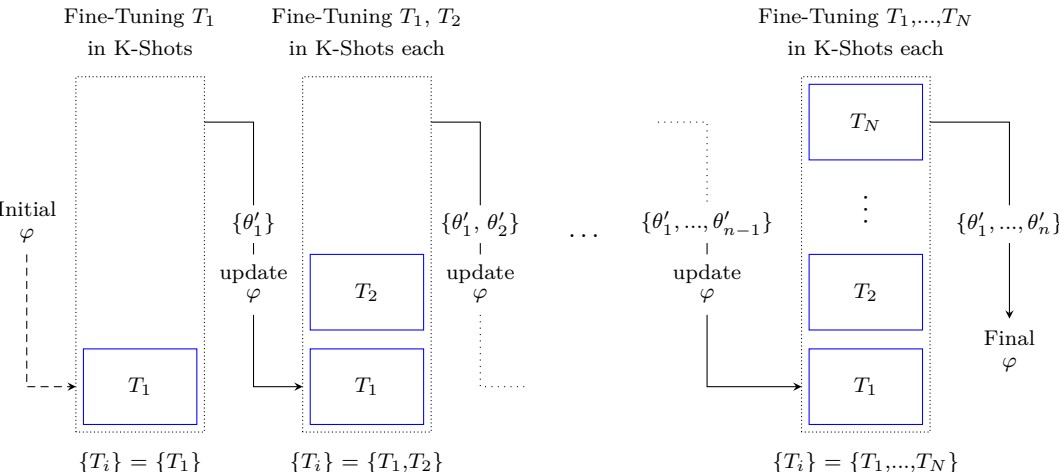

Figure 2: General Meta-Training. In each iteration a new task $T_i$ is sampled from the family $p(T)$. The meta-variable $\varphi$ is the prior for individual $K$ shot fine-tuning of each task. The resulting parameters $\theta'_i$ of each task are used to update $\varphi$.

although it is not a necessary condition, and some works specifically focus on domain generalization Li et al. (2018), Triantafillou et al. (2020).

In the task-based paradigm, there are two components of what must be learned from a theoretical point of view: Common knowledge over all tasks $T_i$ in $p(T)$, and task-specific knowledge gained through few-shot fine-tuning. Hence, the meta-learning paradigm consists of two stages, the more abstract meta-task-level and the few-shot standard learning of a particular task $T_i \sim p(T)$. This paradigm is explicitly applied to all gradient-based meta-learning algorithms presented in Section 3.1. However, even for the memory-based meta-learners presented in Sections 3.2 and 3.3 that do not explicitly divide learning into meta-learning and task-specific standard learning, the meta-training paradigm can be formalized as (implicitly) two-staged by introducing a meta-variable $\varphi$ encoding common knowledge over all tasks. As a consequence, the training-validation-test split is as well two-staged. On meta-task level, certain tasks get explicitly excluded from the meta-training task pool to function as validation throughout meta-training or as test tasks after meta-training, respectively. The corresponding evaluation takes place on the stage of task-specific standard learning, which is why it is often referred to as inner learning. However, since every task $T_i$ from the distribution $p(T)$ is assumed to be a few-shot learning task, a task-specific validation set $X^i_{\text{val}}$ is not required. This way, standard few-shot learning is mimicked within each task $T_i$ while meta-training provides every such task-specific fine-tuning with a meaningful prior to enable fast adaptation.

**Meta-Training and Meta-Testing**

The corresponding meta-training scheme aims to optimize $\varphi$ to be the best prior for fast inner learning. As highlighted in Figure 2, this means to yield task-specific parameters from the general knowledge $\varphi$ and measure their test set performance after $K$ shots of task-level adaptation. The corresponding optimization problem in each meta-training iteration is

$$\text{Minimize} \quad \mathbb{E}_{T_i \sim p(T)} \mathcal{L}^{\text{meta}} \left( \theta^*_{T_i}(\varphi), \ \varphi, \ X^i_{\text{test}} \right) \quad \text{wrt. } \varphi. \tag{3}$$

The notation $\theta_i(\varphi)$ highlights the relationship between $\varphi$ and the initial parameters of the inner learning, while $\mathcal{L}^{\text{meta}}$ denotes the meta loss function and $\theta^*_i(\cdot)$ the optimal task-specific parameters for Task $T_i$ when starting inner learning with meta-parameter $\varphi$. However, the optimal task-specific parameters $\theta^*_i(\cdot)$ are rather a theoretical component of the formalization of meta-training taken from Upadhyay (2023) than a practical implementation.

Inner learning aims to approximate the task-specific optimal parameters $\theta^*_i$ representing the optimal solution of the task $T_i$. Thus, the resulting parameters must be denoted differently to distinguish them from the

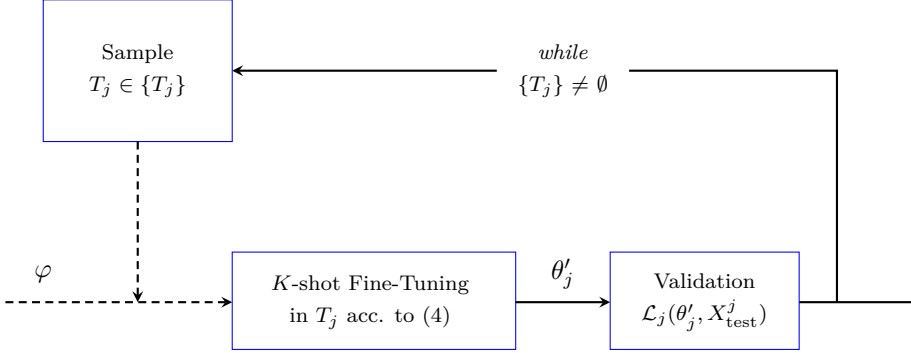

Figure 3: General Meta-Testing Paradigm. For each $T_j$ sampled from the set of test tasks the parameters $\theta_j(\varphi)$ are fine-tuned in $K$ shots, before the resulting $\theta_j'$ get evaluated on the task's test set via the task-specific loss $\mathcal{L}_j$ to yield the performance.

optimal ones. Throughout this work, the respective notation is $\theta_i' := \theta_i'(\varphi)$ for parameters trained in inner learning starting from $\theta_i(\varphi)$ i.e., from initial parameters $\theta$ yielded from the prior $\varphi$. The resulting optimization problem for inner learning is

$$\text{Minimize} \quad \mathcal{L}_i\left(\theta_i, \theta_i(\varphi), X_{\text{train}}^i\right) \quad \text{wrt. } \theta_i. \tag{4}$$

Besides architectural choices, the choice of the meta-loss $\mathcal{L}_{\text{meta}}$ as well as of the inner learning determines the respective meta-learning framework. This holds for all algorithms presented in Section 3, although the meta-loss is not explicitly given for the memory-based meta-learners in Section 3.2. Moreover, the task sampling throughout meta-training is algorithm-specific. Figure 2 shows a scheme where tasks are iteratively drawn from $p(T)$ up to $N$ tasks, but other algorithms might directly start with $N$ tasks and re-sample them, respectively. The numbers $N$, $N_{\text{val}}$ and $N_{\text{test}}$ of training, validation, and testing tasks are themselves hyperparameters of meta-training.

After meta-training a meta-model $f_\varphi$, meta-testing aims to evaluate the trained model on unseen test tasks. However, evaluating the quality of a meta-model means examining how well it boosts task-specific learning. Hence, the task-specific parameters $\theta_j(\varphi)$ are adapted within $K$ shots of inner learning for each test task, while only the meta-variable $\varphi$ is fixed throughout the whole meta-testing process. Evaluating the correspondingly adapted task-level parameters $\theta_j'$ on the task-level test set $X_{\text{test}}^j$ yields the task-level performance required for the meta-performance measures presented in the next section. Figure 3 shows this iterative meta-testing scheme.

**Performance Measures**

In contrast to standard learning, where determining the performance of a learner $f_\theta$ simply means evaluating its loss $\mathcal{L}$ on unseen test data, determining how well a meta-learned prior $\varphi$ boosts fine-tuning any $\theta_j$ to an unseen test task $T_j \sim p(T)$, is more challenging. The mere task-level test set performance is not sufficient to properly evaluate the meta-learning outcome. Instead, various measures must be used to draw a more detailed picture of the adaptation capability the model $f_\varphi$ developed through meta-training. However, in most works, the notion performance refers to the total loss collected within the task-level test batches, if not stated otherwise, while the notion "asymptotic performance" refers to the accumulated loss gained after a full, standard training. [3] The following paragraphs define, motivate, and discuss the most important performance measures used for evaluating meta-learning results.

**Generalization:** In standard learning, generalization refers to the ability of a learner to perform well on unseen data after training. It is generally quantified by the generalization error, i.e., the difference between the learner's performance on the training set and its performance on a test set:

$$\mathcal{L}_{\text{gen}}(\theta) := \mathcal{L}(\theta, X_{\text{test}}) - \mathcal{L}(\theta, X_{\text{train}}). \tag{5}$$

---

[3] The notion "asymptotic" implies $K \to \infty$, which, from a theoretical point of view, is what happens in standard learning.

Analogously, the generalization ability of a meta-learner $f_\varphi$ can be quantified by the accumulated generalization error over the unseen meta-test tasks:

$$\mathcal{L}_{\text{gen}}^{\text{acc}}(\varphi) := \mathcal{L}_{\text{test}}^{\text{meta}}(\varphi) - \mathcal{L}_{\text{train}}^{\text{meta}}(\varphi), \tag{6}$$

where the accumulated train and test losses are defined as the sum over the respective (standard learning) train and test losses

$$\mathcal{L}_{\text{train}}^{\text{meta}}(\cdot) := \sum_{T_j \sim p(T)} \mathcal{L}(\cdot, X_{\text{train}}^j), \qquad \mathcal{L}_{\text{test}}^{\text{meta}}(\cdot) := \sum_{T_j \sim p(T)} \mathcal{L}(\cdot, X_{\text{test}}^j),$$

and $\theta'_K(\varphi)$ denotes the adapted parameters after $k$ training shots on the respective task $T_j$ starting from prior $\varphi$. The accumulated generalization error (6) measures the generalization performance at task level. Instead, one measures meta-generalization by abstracting (6) to the meta-generalization error

$$\mathcal{L}_{\text{gen}}^{\text{meta}}(\varphi) := \mathcal{L}_{\text{gen}}^{\text{test}}(\varphi) - \mathcal{L}_{\text{gen}}^{\text{train}}(\varphi), \tag{7}$$

where $\mathcal{L}_{\text{gen}}^{\text{train}}$ and $\mathcal{L}_{\text{gen}}^{\text{test}}$ denote the accumulated generalization error (6) summed over all meta-training tasks or meta-test tasks, respectively. The meta-generalization error evaluates how well the model $f_\varphi$ generalizes on the meta-level by taking its task-specific generalization capability on the training tasks into account. A good meta-generalization means that $\varphi$ boosts task-specific learning equally well for training and test tasks, while a bad meta-generalization hints $\varphi$ to overfit to the training tasks in such a way, that fine-tuning has the maximal success, but this knowledge cannot be transferred to other tasks from the same distribution. We define the latter as meta-overfitting. However, it is a rather theoretical concept, as the iterative sampling of training tasks throughout meta-training should avoid such an issue.

**Adaptation speed:** Adaptation speed refers to the rate at which a learner can effectively learn new tasks $T_j$ from a limited number of examples. It is typically measured by tracking task-specific metrics over time and examining the slope of the gained curve. A high slope indicates fast adaptation, and vice versa. In the context of meta-learning, one typically tracks adaptation performance during the individual training of each test task $T_j$, for example by the task-specific training loss $\mathcal{L}_{\text{train}}^j$. [4] This way, one yields an adaptation performance measure $\mu_{\varphi, T_j}^{\text{adapt}} : K \to \mathbb{R}$ with

$$\mu_{\varphi, T_j}^{\text{adapt}}(k) := \mathcal{L}_{\text{train}}^j(\theta_j^{(k)}, X_{\text{train}}^{j,(k)}) \tag{8}$$

for each task $T_j$, where $(k)$ denotes the parameters $\theta_j(\varphi)$ after $k$ epochs of inner learning. Calculating (8) for each test task $T_j$ and each fine-tuning iteration $k$ yields the meta-adaptation performance.

**Sample-Efficiency:** Since gathering data is often difficult or expensive in the context of few-shot learning, it is also desirable to yield meta-learners that adapt to unseen situations without the need for large amounts of data. Such learners are referred to as sample-efficient. In contrast to adaptation speed, which measures how fast a model can learn and improve within a certain amount of training iterations regardless of the samples needed, sample-efficiency is about learning effectively from fewer examples regardless of the training iterations required for extracting the data's information. Analogously to the adaptation speed (8), it is measured by [5]

$$\mu_{\varphi, T_j}^{\text{sample eff}}(S) := \mathcal{L}_{\text{train}}^j(\theta_j^{(S)}, X_{X \subseteq X_{\text{train}}^j}^{|X|=S}), \tag{9}$$

where $\theta_j^{(S)}$ denotes the parameters $\theta_j(\varphi)$ fine-tuned with $S$ observations.

**Out-of-distribution:** While generalization refers to a learner's ability to perform well on unseen data that is drawn from the same distribution as the training data. Out-of-distribution (OOD) performance specifically evaluates how well a model can handle data that comes from a different distribution. In other

---

[4]Note that, in theory, any task-specific loss can be used for evaluating the adaptation speed. For example, using the task-specific generalization (5) one examines, how fast a model gains generalization capabilities within one particular task $T_j$.

[5]Note that, similar to adaptation speed, sample-efficiency can be measured with any kind of task-specific loss.

words, while generalization measures how well the learner can apply learned patterns to new examples within the same context, OOD assesses a learner's ability to generalize to new, unseen situations not represented during training. Consequentially, high OOD performance indicates robustness and adaptability, while poor performance can lead to failures in real-world applications where conditions may vary significantly from the training scenarios. On meta-level, this means to define the test tasks as the family $\{T_j\}_{T_j \nsim p(T)}$ of tasks particularly not drawn from the task distribution $p(T)$. Then, measuring the performance means calculating all the metrics mentioned above on these OOD tasks.

## 2.2 Meta-Reinforcement Learning

This subsection transfers the general meta-learning paradigm to that of meta-Reinforcement Learning (meta-RL). Mimicking the structure of Section 2.1, it starts with presenting the standard reinforcement learning (RL) paradigm and thereafter derives the meta-RL paradigm from it.

### Standard Reinforcement Learning

Standard RL is a special case of the standard machine learning paradigm, where an agent interacts with an environment $(\mathbb{A}, \mathbb{S}, R)$ by selecting an action $a \in \mathbb{A}$ based on the state $s \in \mathbb{S}$ and receiving the next state $s' \in \mathbb{S}$ and a reward signal $r \in \mathbb{R}$. Since the agent's actions have a direct effect on the subsequent state, the transition of a general task (1) extends to a distribution $\mathbb{T} : \mathbb{S} \times \mathbb{A} \times \mathbb{S} \to \mathbb{R}$ of subsequent states given a state and a taken action. The reward function $R : \mathbb{A} \times \mathbb{S} \to \mathbb{R}$ and the corresponding discount factor $\gamma \in [0, 1]$ determine the task loss $\mathcal{L}$. In RL, both these functions must fullfill the Markov property so that the standard task (1) forms a Markov Decision Process (MDP)

$$M := (\mathbb{A}, \mathbb{S}, R, \gamma, \mathbb{T}, \mu, h), \tag{10}$$

with action space $\mathbb{A}$, state space $\mathbb{S}$, episode horizon $h$ [6] and initial state distribution $\mu$. Correspondingly, in a RL task, the goal is to find an action selection strategy $\pi : \mathbb{S} \to \mathbb{A}$ that maximizes the reward collected throughout an episode of agent-environment interaction in the corresponding MDP $M$. This yields the value function [7]

$$\mathcal{L} := -\mathbb{E}_{\pi_\theta(\cdot)}^{s_0 \sim \mu} \sum_{t=0}^{h} \gamma^t r_{t+1} \tag{11}$$

where $r_{t+1} = R(s_t, a_t)$ and $\mathbb{E}_{\pi_\theta(\cdot)}^{s_0 \sim \mu}$ denotes the expectation under the assumptions that $s_0$ is drawn from the initial distribution $\mu$ and every action $a_t$ is taken according to policy $\pi_\theta(s_t)$. Since a RL task $T_i$ is determined by the underlying MDP $M_i$, this work uses the notions task and MDP synonymously.

There are almost as many rules for updating $\theta$ as RL algorithms, and not all of them explicitly minimize (11) Sutton & Barto (2018), Francois-Lavet et al. (2018). For example, The family of function approximation RL algorithms aims to approximate the loss (11) (or modifications of it) by optimizing $\theta$ to best approximate the loss. Afterwards, they achieve the loss minimization by acting accordingly.

---

[6] RL problems do not have to be episodic, but $h < \infty$ is a rather practical condition.

[7] The agent does not neccessarily have to only update its parameters at the end of an episode, although this is assumed throughout this paper. However, generalizing the value function to the expected future reward starting from any state $s_t$ at any time $t$ only requires for an index shift within the sum.

**Example 2 - Racing games:**

Considering a racing game, where the agent's goal is to drive in a way that finishes a racing track as fast as possible, the different components of the underlying MDP (10) are defined as follows:

- The set of possible actions $\mathbb{A}$ contains all directions the racer can move to, its speed and whether or not to brake.

- The state space $\mathbb{S}$ contains all states, the racer can possibly find itself in. Additional to the current racing track (as pixels), it may include information on the racer's speed as well as the time since the start of the race or other useful information a player might have.

- The initial distribution over states $\mu$ refers to what the racer "sees" in its starting position. If that position is deterministic, $\mu$ is deterministic, too.

- The reward function can easily be modeled as one when reaching the goal and zero otherwise.

- The discount factor $\gamma$ must be smaller than one to encourage the agent to finish the racing track as fast as possible.

- Each race is an episode. Hence, the time the agent needs to finish the racing track, determines the horizon $h$ that is therefore dynamic.

The strategy $\pi$ is the theoretical decision rule of the agent. It is representative of the agent as it determines, how the racer actually drives at any time in the race. Since this work only considers deep RL, the policy $\pi_\theta$ is defined by the agent's neural network weights $\theta$.

To learn the optimal strategy $\pi^*$ for a particular racing track, an agent must explore the environment to uncover new actions and their potential rewards, (e.g., it must try out different ways to drive). At the same time, it must exploit its gained knowledge to maximize immediate rewards (e.g., finish the race faster). This trade-off is called the exploration-exploitation dilemma. It particularly motivates the Bayesian RL paradigm that is presented at the end of this section.

Interacting with an environment can be extremely financially expensive e.g., in robotics, trading or transportation. For such problems, an agent must be trained within a simulation and, thereafter, adapt as fast as possible to the real world. Moreover, environmental dynamics often change over time e.g., when shocks permanently increase market prices, when a robot is assigned to a different task within the same physical environment or when a racing track becomes more slippery due to heavy rain. In such scenarios, a standard RL agent must often be trained fully from scratch, although it has already gained a lot of knowledge about the environment that is still valid.

For these reasons, Meta-RL aims to collect general knowledge over a family of similar MDPs. The goal is the best possible fast adaptation to any MDP from that family, i.e., adapting to any MDP within only $K$ episodes. One can easily modify Example 2 into such a meta-RL problem by parameterizing the racing track's slipperiness. For each different value of slipperiness the transition $\mathbb{T}$ of the underlying MDP slightly changes, while the overall structure of state and action spaces and even reward remain unchanged. Such families of MDPs are called parametric. Acting optimally in any of these environments requires almost the same high-level skills: The agent still needs to learn driving dynamics, when to brake and which direction to drive to, just with a different level of drifting in curves due to a change in slipperiness. In other words, the meta-problem to finish a randomly slippery racing track as fast as possible, remains the same. The following paragraphs derive the meta-RL paradigm from meta-learning, while Figure 4 applies it to the meta-racing-problem accordingly.

**The Meta-Reinforcement Learning Paradigm**

Analogous to meta-learning, the meta-RL paradigm formally consists of a distribution $p(M)$ over MDPs of the form

$$M_i := (\mathbb{S}, \mathbb{A}, R_i, \mathbb{T}_i, \mu_i, \gamma, h). \tag{12}$$

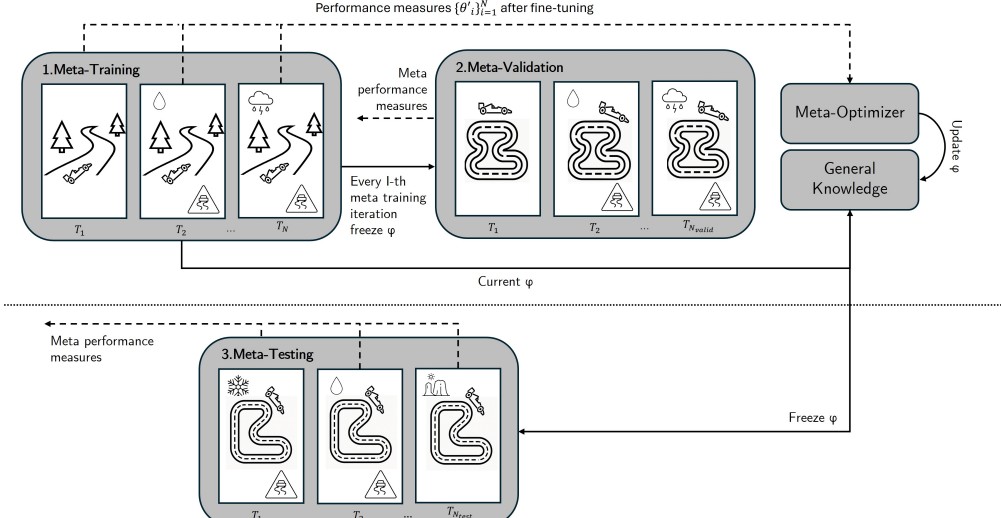

Figure 4: Meta-learn to race on tracks with varying weather conditions. Starting from the meta-knowledge $\varphi$, $K$ episodes of fine-tuning on the particular racing track yield performance measures. The meta-optimizer uses these measures to update meta-knowledge $\varphi$. Meta-validation evaluates the training progress on unseen racing tracks every $l$ meta-epochs. After meta-training, the meta-policy adapts to test tracks to evaluate the quality of prior $\varphi$.

The MDPs from the same family $\{M_i\}_{M_i \sim p(M)}$ normally vary with respect to their underlying dynamics $\mathbb{T}_i$ or reward function $R_i$, while sharing some structure like state and action space - which, together, can be interpreted as the observation space $\mathcal{X}$ of a general task (2). However, these assumptions are rather practical (see e.g., Melo (2022) or Duan et al. (2016)) than an actual theoretical necessity.

Collecting knowledge within a family of MDPs (possibly) means two similar but different things:

1. Collecting general knowledge about structures like state or action space that are shared within the family (see sections 3.2.1 and 3.3.1), or

2. Identifying the current MDP as fast as possible to adjust the agent's policy $\pi_\theta$ accordingly (see sections 3.1.2 and 3.2.2).

For racing MDPs of the form presented in Example 2 the former means to learn how to process the pixels shown as the current state, remember general information like the route of the race, and to understand how to drive a car on an abstract level. The latter corresponds to the exploration/exploitation dilemma on MDP-level. It requires figuring out as fast as possible how slippery the racing track actually is in order to adjust the style of driving accordingly. As a consequence, meta-RL is also two-staged: On meta-level common knowledge over all MDPs is collected, while on MDP-level the current MDP is identified to adjust the MDP-specific policy, respectively. The training-validation-test split is analogous to that of meta-learning.

**Meta-Training and Meta-Testing**

Both the inner optimization (4) as well as the outer optimization (3) are straightforward to adjust to the meta-RL training scheme by introducing a meta-variable $\varphi$ representing meta-knowledge. The only core structural difference is in the MDP-specific training and testing data $X_{\text{train}}^i$ and $X_{\text{test}}^i$: As the agent has to interact with its environment to observe the subsequent state and reward, sampling an observation set $X \in \mathcal{X}$ always depends on the current policy. One rather speaks of "collecting experience" of the form $X = \{s_t, a_t, r_{t+1}, s_{t+1}\}_{t=0}^h$. Consequentially, in the meta-RL paradigm, gathering data from a particular MDP $M_i$, in $K$ shots, means collecting $K$ episodes of experience within the environment following the strategy $\pi_{\theta_i(\varphi)}$ that might or might not change between episodes depending on the inner RL algorithm used.

The meta-testing scheme in meta-RL is also analogous to the general meta-testing scheme. The only major difference is in the MDP-level testing of the adapted policy $\pi'_{\theta_j}$ as it interacts with its environment for another $K_{\text{test}}$ episodes after inner learning without further adaptation. The accumulated reward gained throughout these test episodes represents the test loss $\mathcal{L}^j_{\text{test}}$ required for calculating the performance measures presented in the previous section.

**Bayesian Reinforcement Learning**

Bayesian Reinforcement Learning (BRL) provides a sophisticated approach to tackle the exploration/exploitation dilemma by allowing agents to quantify uncertainty. It is the groundwork for the task-representation algorithms presented in sections 3.1.2 and 3.2.2. The main idea of BRL is to leverage the gathered information of the agent together with Bayesian methods in order to enable the agent to adapt its strategy as more information becomes available. For this purpose, one extends the notion of a MDP of the form (10) to that of a Bayes-Adaptive MDP (BAMDP), [8] i.e., a tuple

$$M' := (\mathbb{A}, \mathbb{S}', R', \gamma, \mathbb{T}', \mu', h'), \tag{13}$$

with a hyperstate-space $\mathbb{S}' = \mathbb{S} \times B$ as the extension of the original state space $\mathbb{S}$ by the belief space $B$ capturing the possible parameters of a model of the environment dynamics $\mathbb{T}$ and $R$. [9] The augmented transition and reward functions, as well as the initial distribution and horizon, are defined analogously to the hyper-state space (see Zintgraf et al. (2020), Ghavamzadeh et al. (2015) for further details). In terms of Example 2, this means, for example, to capture a belief of the current slipperiness of the track while collecting more experience to make this belief more precise.

The current belief $b_t := p_t = p(R', \mathbb{T}'|c_t) \in B$ is defined as the posterior of the dynamics model parameters given the current experience (also named context)

$$c_t := \{s_i, a_i, r_{i+1}, s_{i+1}\}_{i=0}^t \tag{14}$$

and a prior distribution $p_0 = p(R', \mathbb{T}')$ over the unknown reward and transition functions $\mathbb{T}$ and $R$. The prior is often chosen as a normal distribution, while one computes the posterior $p_t$ according to the Bayesian rule after collecting experience $c_t$:

$$p_t = p(R, T|c_t) = \frac{p(c_t|R, T)p(R, T)}{\int p(c_t|x)p(x)dx}. \tag{15}$$

The posterior encodes the agent's uncertainty about model environment parameters. However, its computation is generally intractable so that the respective BRL algorithm must approximate it accordingly. Since the agent aims to select the best possible action in each time step based on the current belief, while simultaneously refining the posterior belief whenever new information becomes available, it implicitly trades-off exploration and exploitation this way.

One calls agents Bayes-optimal, if they optimally trade-off exploration and exploitation. Throughout early learning, the value (11) of the corresponding Bayes-optimal policy is typically lower than that of the optimal policy $\pi^*$, since the agent must, at first, explore the environment before finding the optimal dynamics model parameters. In Example 2, the Bayes-optimal behavior initially requires driving and braking a few times to figure out how slippery the track is, although this increases the racing time in the first training episode.

## 3   The Timeline of meta-Reinforcement Learning Landmarks

This section presents the evolution of the meta-RL algorithms leading to Deep Mind's Adaptive Agent (ADA), starting with the class of gradient-based meta-learners. It introduces a new concept with each landmark, following the previously established paradigm as a guiding framework.

---

[8]In Ghavamzadeh et al. (2015) it is done for discrete state and action spaces, but it is straight forward to adapt this for continuous spaces

[9]As all BRL-based algorithms presented in Section 3 use model-based approaches, model-free BRL is not discussed in this work.

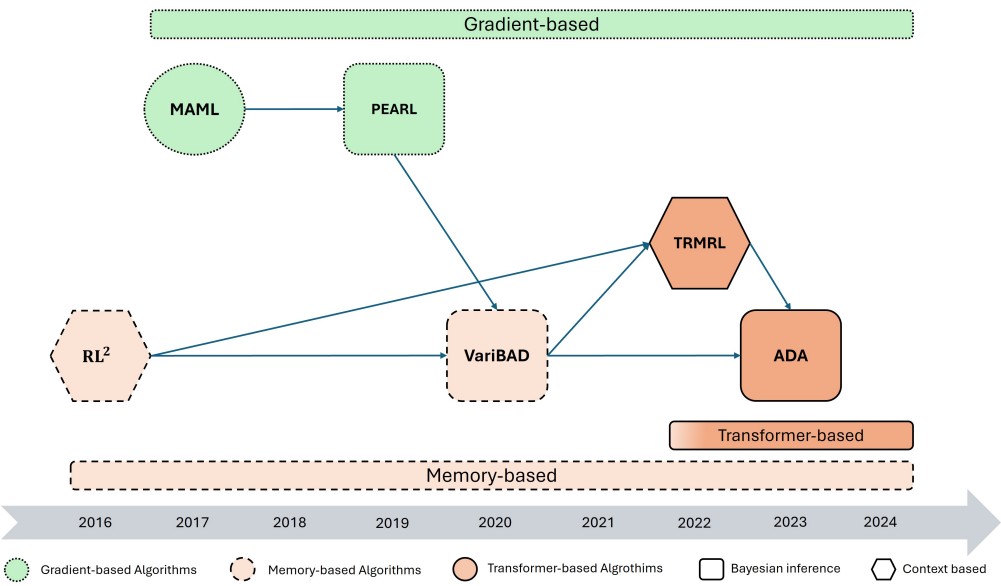

Figure 5: Timeline of landmark developments towards the Adaptive Agent (ADA).

Thereby, the broader family of memory-based (blackbox) meta-RL algorithms distributes into two subsections about RNN-based (Section 3.2) and transformer-based (Section 3.3) methods. The section closes with the current state of the art meta-RL algorithm, ADA, a generalist, transformer-based agent incorporating many common techniques for boosting meta-training such as distillation and auto-curriculum learning. Figure 5 visualizes this timeline, while Tables 1 and 2 compare the different landmarks to each other.

## 3.1 Gradient-based Meta-Learning

Deep neural networks - are gradient-based models, i.e., they are optimized via stochastic gradient descent (SGD) or similar variants of gradient-based optimizers like Adam Kingma & Ba (2017) or AdamW Loshchilov & Hutter (2018). However, these optimizers often converge slowly or not even converge at all. The convergence highly depends - among other factors - on the choice of the starting point of optimization. An optimizer might find the global optimum very quickly when starting the optimization relatively close to it, but is very likely to need many more iterations or even end up in a poor local minimum when starting with randomly initialized model parameters. Several initialization schemes try to adress this issue (e.g., the Glorot Glorot et al. (2011), He He et al. (2015), and LeCun LeCun et al. (1989) initializations). A second major issue is to choose the optimizer's learning rate $\alpha$. Setting it too high can cause the optimizer to be unstable or alternate around some solution, while chosen too low it significantly slows down learning Nar & Sastry (2018).

Both these problems could be tackled, if it were possible to a priori place the parameters in a sufficiently narrow area around the desired optimum. Not only would learning be more stable and directed towards that optimum, but a small learning rate would also suffice to approach it within a few steps. This is the main idea of gradient-based meta-learning.

### 3.1.1 Model-Agnostic Meta-Learning

Model-Agnostic Meta-Learning (MAML) Finn et al. (2017a) combines the meta-learning paradigm with the idea of a priori placing the parameters $\theta$ in an area around the optimal solution $\theta^*$. It is the foundational gradient-based meta-learning algorithm and one of the earliest and simplest landmarks in the evolution of meta-learning and meta-RL. As long as the loss function is sufficiently smooth to compute gradients, it is broadly applicable to any machine-learning problem. As a consequence, many works have successfully adapted the MAML paradigm to various machine learning domains such as neural architecture search Wang

Table 1: Advantages and disadvantages of gradient-based meta-RL landmarks. The last column outlines the problem settings in which each algorithm is most appropriately applied.

| Algo | Advantages | Drawbacks | Use if |
|---|---|---|---|
| MAML | • Usable for meta-learning and meta-RL
• Easy to implement
• Many open-source repos available
• First-order implementations available | • Only PG methods for meta-RL
• Only simple task distributions
• Bad OOD performance | • When minimal development effort is desired
• When the problem's task distribution is simple
• Requires second-order gradients |
| PEARL | • Includes context for task identification
• Off-Policy meta-RL
• More sample-efficient | • More complicated to implement
• Less wider options for open-source code
• Weeknesses in more complex MDP distributions and in tasks with sparse reward | • If sample efficiency and task identification is more important than development time, computation power and simplicity |

et al. (2022a), regression Sen & Chakraborty (2024), multi-object tracking Chen & Deng (2024), time-series classification Wang et al. (2024a), and semi-supervised learning Boney & Ilin (2018), while applying it to various fields like medicine Tian (2024), Alsaleh et al. (2024), Tian et al. (2024), Ranaweera & Pathirana (2024), Naren et al. (2021), biomass energy production Zhang et al. (2025), or fault diagnoses in bearings Lin et al. (2023).

Various works examined MAML's properties. Assuming that MAML's model architecture consists of a sufficiently deep neural network with ReLU activations and that the loss function is either cross-entropy or mean squared error (MSE), Finn & Levine (2018) prove that MAML serves as an universal function approximator for any training set $X_{\text{train}}^i$ and test set $X_{\text{test}}^i$ within an arbitrary task $T_i$. They show, furthermore, that MAML converges linearly to a global optimum under MSE loss when using an over-parameterized deep neural network along with a SGD optimizer like Adam Kingma & Ba (2017). This implies that a deeper model, with at least one hidden layer, is necessary, even if single tasks can be addressed using a shallower or linear model, as demonstrated by Arnold et al. (2021). Empirical findings support that MAML's meta-training typically achieves almost zero training loss and 100% training accuracy (indicating global convergence) when appropriate hyperparameters are utilized Wang et al. (2022a). However, Raghu et al. (2020) note that it leads to effective feature representations rather than a rapidly adaptable prior. During task-specific training, the initial layers of the underlying network exhibit minimal changes, suggesting that the fundamental feature representations remain stable.

Since MAML's universal approximation theorem Finn & Levine (2018) requires the size of the underlying neural network to scale with the number $n$ of tasks, the number of different tasks to be learned simultaneously in the MAML paradigm is naturally bounded by the fact that the model size cannot be increased indefinitely. Additionally, the time required for learning the tasks scales with their complexity Parisotto et al. (2019) as more complex tasks require better exploration and exploitation. Moreover, the MAML paradigm assumes the optimal parameters $\theta_1^*, \theta_2^*, \ldots$ of the different tasks $T_1, T_2, \ldots$ to be sufficiently close to each other, so that a general prior $\varphi$ (i.e., a global optimum) can be found from which each optimal solution can be sufficiently well approximated within $K$ gradient descent steps. But as parameter vectors are likely to be high-dimensional, this might be a rather strong assumption. These drawbacks explain why the memory-based baseline algorithms presented in Sections 3.2 and 3.3 outperform MAML, especially in terms of generalization and OOD performance. They, moreover, provide the main motivation for the development of MAML extensions such as Robust MAML Nguyen et al. (2021) and XB-MAML Lee & Yoon (2024), which are specifically designed to improve generalization to tasks that are slightly out of distribution."

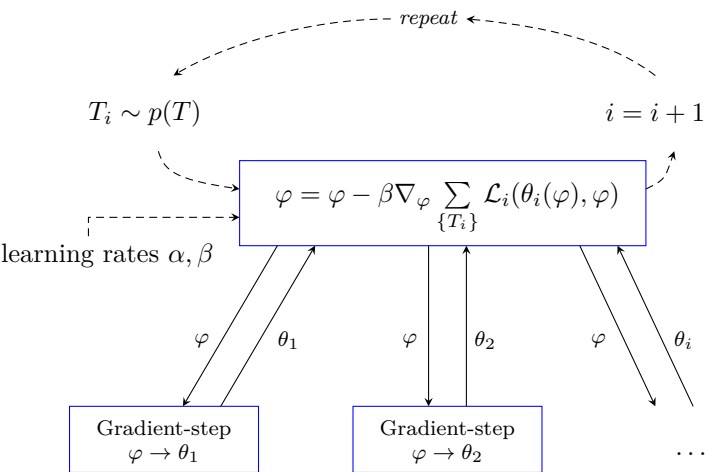

Figure 6: The MAML meta-training scheme.

As the original MAML paradigm relies of second-order gradients that are usually computationally expensive, Finn et al. (2017a) suggested First-Order (FO) MAML, a more efficient approach that omits second-order derivatives. Although this modification does not guarantee global convergence Fallah et al. (2020) (a limitation that motivated further modifications like Hessianfree MAML Fallah et al. (2020) or Implicit MAML Rajeswaran et al. (2019)), MAML in its first-order approximation is much easier to implement and computationally less expensive than memory-based meta-learning algorithms presented in the subsequent sections Finn et al. (2017a).

**MAML Paradigm**

The original MAML paradigm directly derives from the general meta-learning paradigm presented in Section 2.1. On the meta-level, the main idea is to place the prior $\varphi$ in a region that is promising for all tasks drawn from the distribution $p(T)$, so that starting each task-specific fine-tuning from that prior, i.e., setting $\theta_i(\varphi) = \varphi$ for all $\theta_i$, $K$ gradient descent steps during fine-tuning maximize task-specific performance.

**Meta-Training**

As the meta-training of MAML directly derives from the general meta-training scheme presented in Section 2.1, it consists of a meta-level and a task-specific stage. On meta-level, the meta-loss $\mathcal{L}_{\text{meta}}$ is defined as the sum over all task-specific losses $\mathcal{L}_i$ on the respective test sets $X_{\text{test}}^i$:

$$\mathcal{L}_{\text{meta}} := \sum_{i=1}^{N} \mathcal{L}_i\big(\big(\theta_i',\ \varphi,\ X_{\text{test}}^i\big)$$

with $\theta_i'$ denoting the task-specific parameters resulting from the individual inner update steps (17). the meta-level gradient descent update of $\varphi$ uses the performance of the updated parameters $\theta_i'$ on the respective test set $X_{\text{test}}^i$ to optimize $\varphi$ with respect to the inner learning performance. For $K = 1$ this is

$$\varphi' = \varphi - \beta \nabla_\varphi \mathcal{L}_{\text{meta}} \tag{16}$$

with pre-defined meta-learning rate $\beta$.

At task-level, the inner optimization problem (4) is also solved by $K$ gradient descent steps of the form

$$\theta_i' = \varphi - \alpha \nabla_\varphi \mathcal{L}_i\left(\varphi, X_{\text{train}}^i\right), \tag{17}$$

with a pre-defined learning rate $\alpha$, [10] and the meta-level parameters $\varphi$ as the corresponding task-level prior. The loss $\mathcal{L}_i$ depends on the specific task and its training set $X_{\text{train}}^i$. One typically assumes it to be calculated with all $K$ examples $x \in X_{\text{train}}^i$ at once, although this is no algorithmic necessity.

Figure 6 shows one iteration of the MAML meta-training scheme with $K = 1$. However, as the meta-testing of MAML fully follows the scheme presented in Figure 3, no additional figures shows the MAML meta-testing scheme.

**Meta-Reinforcement Learning**

Adjusting MAML to the meta-RL paradigm is straightforward. Besides the fact that task-specific training and test sets must be represented by $K$ episodes of MDP-specific agent-environment interaction, the general MAML structure of embedding task-specific gradients in a meta-gradient descent scheme remains the same. However, the gradient descent steps (17) and (16) require gradients of the task-specific losses, which are the task-specific expected returns (11) in meta-RL. But these value functions are not differentiable due to unknown environment dynamics and the agent's impact on them. Consequentially, the original MAML meta-RL paradigm can only apply policy-gradient methods Sutton et al. (1999), what makes MAML on-policy. The exact computation of the policy gradient depends on the respective task-level algorithm. However, many state-of-the-art standard RL algorithms, such as PPO Schulman et al. (2017) or TRPO Schulman (2015), utilize an off-policy learning scheme to learn from many policies at once. This motivates the PEARL Rakelly et al. (2019) algorithm, the landmark off-policy modification of MAML, which is introduced in the next subsection.

The meta-update of $\varphi$ requires for policy gradients $\nabla_\varphi \pi_{\theta_i'}$ of the updated parameters $\theta_i'$ on the respective task-specific test episodes $X_{\text{test}}^i$. In this way, the underlying policy-gradient algorithm is extended to the meta-level, i.e. to updating $\varphi$ with respect to the performance of the fine-tuned parameters $\theta_i'$ on test sets $X_{\text{test}}^i$. However, this meta-update scheme consists of second-order terms: The test set policy gradients $\nabla_\varphi \pi_{\theta_i'} \mathcal{L}_i X_{\text{test}}^i$ result from train set policy gradients $\nabla_\varphi \pi_\varphi \mathcal{L}_i X_{\text{train}}^i$. These second-order gradients are difficult to derive, what motivates meta-RL-specific first-order MAML extensions such as Taming-MAML Liu et al. (2019) or DICE Foerster et al. (2018).

### 3.1.2 Efficient Off-Policy Meta-RL

The identification of the current task $T_i$ i.e., the hidden dynamics of the corresponding MDP $M_i$, plays a crucial role in learning the task-specific optimal behavior $\pi_{\theta_i^*}$. During MDP-level learning, this identification can only rely on the experiences collected in the particular MDP so far. Such experience is called context and numerous meta-learning algorithms incorporate it into their paradigm Zintgraf et al. (2019), Ni et al. (2022), Ben-Iwhiwhu et al. (2022), Zhang et al. (2021), Fu et al. (2021), Deng et al. (2024). The better the agent explores its environment, the more informative the context becomes. In Example 2, the current context is defined by all segments of the racing track the racer has visited so far. The more of the track it explores, the more informative the context becomes, and thus the easier the current track can be fully identified.

Additionally, there is uncertainty about which MDP the agent is currently acting in. The actions taken by the current agent determine, which experiences it gathers, but unseen states might be more informative about the current MDP and hence other actions might have been better for exploration. This uncertainty can be incorporated into the meta-RL framework by utilizing the Bayesian Reinforcement Learning paradigm introduced at the end of Section 2.2. Thereby, the context a posteriori updates the belief over the current MDP $M_i$, while the corresponding posterior distribution informs the decisions of the agent. Algorithms following this principle are referred to as task-inference methods (see, e.g., Beck et al. (2023b)), whereas algorithms like CAVIA Zintgraf et al. (2019) that utilize the current context without Bayesian inference are called context-based. Task-inference methods appear both in the family of gradient-based meta-learners Ni et al. (2022), Ben-Iwhiwhu et al. (2022), Gupta et al. (2018), Zhang et al. (2021), and in the broader family of memory-based approaches Zintgraf et al. (2020), Bing et al. (2024).

---

[10]Finn et al. (2017a) state that the learning rate $\alpha$ can also be meta-learned, what is adressed in $\alpha$-MAML Behl et al. (2019)

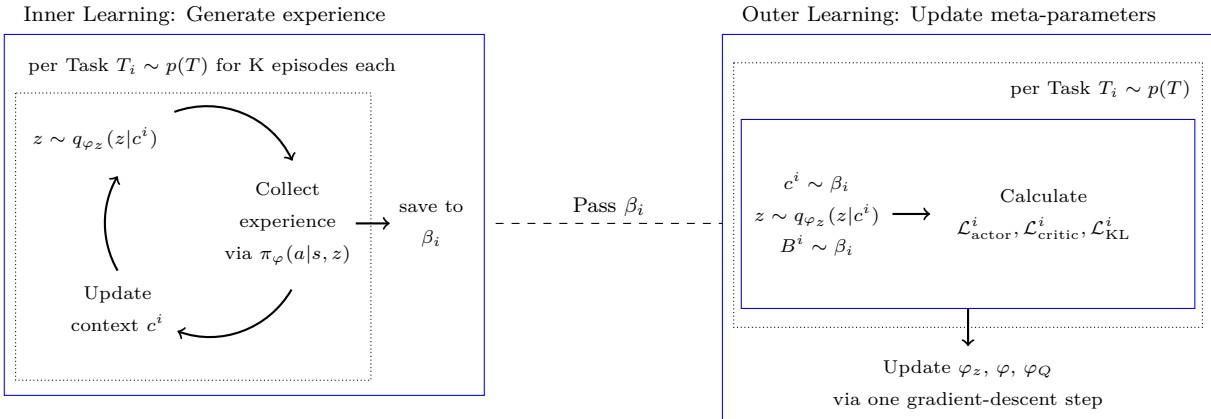

Figure 7: The PEARL meta-training scheme. In each MDP $M_I$, PEARL collects $K$ episodes of experience and stores them in replay buffer $\beta_i$. It samples the context variable $Z$ at the beginning of each episode. After inner learning, minibatches are sampled from the replay buffer to update meta-parameters $\varphi, \theta_\pi$ and $\theta_Q$.

Efficient Off-Policy Meta-RL (PEARL) Rakelly et al. (2019) is such a Bayesian Inference meta-RL algorithm. It outperforms MAML as well as the simplest memory-based landmark $RL^2$ (see Section 3.2.1) in terms of few-shot performance, adaptation speed and sample-efficiency on the Mujoco benchmark Todorov et al. (2012). Hence, it is a frequently used benchmark for other meta-RL algorithms and the gradient-based task inference landmark on the timeline presented in this section. In contrast to other extensions of MAML, such as $\alpha$-MAML Behl et al. (2019) or PMAML Finn et al. (2018), which only slightly modify the original paradigm, PEARL introduces a substantially different meta-learning scheme. It not only incorporates task inference (as also proposed in Finn et al. (2018) or Grant et al. (2018)), but also reformulates the meta-training scheme to off-policy learning. This modification is likely to be the main reason for PEARL's superior adaptation speed and sample efficiency. Nevertheless, PEARL shows weaknesses in more complex MDP distributions and in tasks with sparse rewards. This motivates the more recent and sophisticated gradient-based algorithms DREAM Liu et al. (2021), MAESN Gupta et al. (2018) and MetaCure Zhang et al. (2021) that are more tailored to optimally explore at the beginning of MDP-specific adaptation. However, the following paragraphs examine the PEARL paradigm and training scheme from which these modifications can easily be derived.

**The PEARL Paradigm**

The main idea of PEARL is to infer a MDP-level latent context variable $z_i$ via Bayesian updates that encodes an MDP $M_i$ based on the current context $c_t^i := \{(a_j, s_j, r_j, s_j')\}_{j=1\dots,t}$. This context variable can either represent function approximations like the value function (11) (model-free approach), or the dynamics $\mathbb{T}_i$ and $R_i$ of the current MDP $M_i$ (model-based approach). It is sampled from a prior network $p_\varphi(z_i|c_i)$ at the beginning of each RL episode, whose weights $\varphi_z$ are learned at meta-level. During inner learning, the prior network updates implicitly through the context and thereby approximates the corresponding Bayesian posterior update (15).

The policy $\pi(\cdot|z_i)$ is a neural network with meta-learned weights $\varphi$. According to the BRL paradigm, it depends on the context variable $z_i$, i.e., on the belief about the current MDP. Hence, it implicitly adapts to a particular MDP through the context, so that a MDP-level gradient descent update of the policy weights $\varphi$ is not required. This makes PEARL off-policy: the policy $\pi_\varphi$ is only updated at meta-level, but remains unchanged throughout MDP-level learning. Instead, the belief updates based on the gathered experience. In their original publication, Rakelly et al. (2019) used Soft-Actor-Critic Haarnoja et al. (2018). However, this is rather a design choice than a necessity of the PEARL paradigm, but it incorporates a critic $Q_{\varphi_Q}$ into the framework whose parameters $\varphi_Q$ must also be meta-learned via meta-gradient descent.

**Meta-Training and Meta-Testing**

Figure 7 illustrates the meta-training scheme of PEARL. At the beginning of each RL episode, $z_i$ is sampled from the prior distribution $q_{\varphi_z}(\cdot|c_i)$ conditioned on the current context $c^i$. Then, the policy $\pi_\varphi(\cdot|z_i)$ gathers one RL episode of experience $c_h^i$, and a MDP-specific replay buffer $\beta_i$ stores this experience for meta-level loss calculation. a MDP-specific test set $X_{\text{test}}^i$ is not required, [11] As all meta-losses are calculated on the experience gathered and stored this way.

The meta-level updates of $\varphi$, $\varphi_z$ and $\varphi_Q$ are one meta-gradient descent step of the form (16), with the meta-losses for the actor $\pi_\varphi$ and critic $Q_{\varphi_Q}$ defined as the sum of the respective MDP-level losses. To stabilize training, the meta-loss for $\varphi_z$ contains a KL-divergence penalty

$$\mathcal{L}_{\text{KL}}^i := \text{D}_{\text{KL}}(p(z_i \mid c_t^i), r(z_i))$$

with context $c_t^i$ sampled from replay buffer $\beta_i$. The baseline distribution $r$ is a hyperparameter of PEARL and chosen as a normal distribution in the original publication. The corresponding meta-loss for $\varphi_z$ is

$$\mathcal{L}_{\text{meta}}^{\varphi_z} := \sum_{M_i} \left( \mathcal{L}_{\text{critic}}^i + \mathcal{L}_{\text{KL}}^i \right).$$

It, additionally, contains the MDP-level critic losses, since the functions like (11) frequently used for critics highly depend on which MDP the agent is currently acting in.

The meta-testing scheme of PEARL is analogous to the general meta-testing scheme presented in Section 2.2. However, as the experience gathered during inner learning are saved in the replay buffer, no test episodes are required for performance evaluation. But, similar to meta-training, this is rather a design choice of Rakelly et al. (2019) than a neccessity.

## 3.2   Memory-based Meta-RL

In meta-RL, all MDPs are drawn from a meta-distribution $p(M)$ over MDPs that are similar but distinct. Memorizing similarities facilitates the adaptation to a new MDP that requires a similar strategy. Additionally, RL problems are sequential so that the ability to process sequences instead of single transitions $(s, a, r, s')$ helps to cover time-dependencies and to solve RL problems where rewards are sparse. Altogether, this gives rise to the idea of utilizing memory architectures.

The simplest memory architecture is that of a Recurrent Neural Network (RNN). RNNs maintain a hidden state $h$ that acts like a memory mechanism by storing information about past experiences. Although more recent advances like TRMRL Melo (2022) utilize transformer networks as the more sophisticated memory structure, following the timeline of memory-based meta-RL, this section discusses the landmark RNN-based meta-RL algorithms, starting with RL$^2$ as the earliest and simplest landmark in Section 3.2.1.

RNNs are Turing-complete universal computers Tur. However, even in standard RL, RNN-based agents are very sensitive to architectural choices like their initialization, the choice of the underlying RL algorithm, or the length of the context that is supposed to be captured by the hidden state $h$ Ni et al. (2022). The latter is, moreover, highly MDP-specific Ni et al. (2022), which limits the flexibility of RNN-based meta-RL algorithms and makes them struggle to generalize between tasks Beck et al. (2023a), Ben-Iwhiwhu et al. (2022). This motivates additional modifications like using a hypernetwork Beck et al. (2023a) to learn initializations of the MDP-specific RNN weights, as well as incorporating Bayesian inference in the RNN-based meta-learners. The latter motivates VariBAD, the landmark RNN-based task inference meta-RL algorithm presented in Section 3.2.2.

### 3.2.1   Fast Reinforcement Learning via Slow Reinforcement Learning

The RL$^2$ algorithm Duan et al. (2016) is the earliest landmark presented in this work. As such, several surveys use it as the main reference for black-box meta-RL Beck et al. (2023b), Vettoruzzo et al. (2024),

---

[11]It is rather a design choice than a theoretical necessity to not sample another $K$ episodes with fixed $z_i$ or $p_{\varphi_z}$.

Table 2: Advantages and disadvantages of memory-based meta-RL landmarks. The last column outlines the problem settings in which each algorithm is most appropriately applied.

| Algo | Advantages | Drawbacks | Use if |
|---|---|---|---|
| RL$^2$ | • Simple, foundational approach for memory-based meta-RL
• Bayes-optimal on simple benchmarks | • Limited generalization to complex MDP distributions
• Sensitive to architecture, context length, and RL algorithm choice
• Low sample efficiency and unstable adaptation | • When a conceptually clear baseline is needed
• Suitable for small, well-structured task distributions |
| VariBAD | • Explicitly models task uncertainty
• Can exhibit near Bayes-optimal exploration
• Divides into different components | • Unstable on more complex or out-of-distribution tasks
• Higher architectural and computational complexity
• Bayes-optimality not generally proven | • When sample efficiency and uncertainty modeling are key
• Suitable for tasks requiring rapid exploration and adaptation |
| TRMRL | • Meta-learner per design
• Better sample efficiency, adaptation speed and particularly OOD performance than all other landmarks
• Transformers have a demonstrable long-term memory | • Transformers require vast amount of data and computation resources for (meta-)training
• Performs worse on tasks with high task uncertainty | • Vast amount of training data and power is available
• Very powerful problem solver is requiered
• Task uncertainty is not particularly high |

while it serves as a benchmark for all gradient-based meta-learners presented in the previous section as well as algorithms like VariBAD Zintgraf et al. (2020) or TRMRL Melo (2022) discussed later on. The main idea is to simply combine a standard RL algorithm with an RNN-based agent. This way, the general knowledge $\varphi$ is represented by the hidden state of the RNN. The corresponding RNN weights update only once every $K$ episodes of MDP-specific learning, which aligns with the notion of "slow learning", while the activation of the RNN's hidden states serves as an implicit fast adaptation scheme throughout these $K$ episodes. The originally published algorithm uses TRPO Schulman (2015) as the standard RL algorithm, although Rakelly et al. (2019) show RL$^2$ to perform even better with more sophisticated RL algorithms like Proximal Policy Optimization (PPO) Schulman et al. (2017).

The RL$^2$ algorithm was experimentally shown to be Bayes-optimal on simple benchmark tasks like $N$-arm bandit problems Mikulik et al. (2020). The experiments indicated the Bayes-optimal solution to be a fixpoint of meta-training. However, the task distributions used in Mikulik et al. (2020) were manually constructed so that calculating a bayes-optimal solution was tractable. The performance on broader, more complex task distributions is not guaranteed. In fact, the gradient-based meta-RL algorithms presented in the previous section outperform RL$^2$, which gave rise to more sophisticated methods like VariBAD Zintgraf et al. (2020), whose objective is explicitly tailored to optimize towards Bayes-optimality. The next paragraphs, nevertheless, describe RL$^2$ under the general meta-RL paradigm of Section 2.2 since it is the foundational approach for memory-based meta-RL.

**Paradigm**

RL$^2$ can be differentiated into an inner and an outer learning stage Duan et al. (2016), although there, in fact, is only one particular policy $\pi_\theta$ represented by an RNN interacting with different MDPs through an

environment. General knowledge is implicitly collected by updating the weights of the RNN once every $K$ episodes, while the MDP-specific behavior is only updated implicitly via the model activations following from the currently gained experience. This way, the goal to Bayes-optimally trade-off exploration and exploitation is also implicit: The objective of maximizing rewards over $K$ episodes results in the implicit need to explore the current MDP as fast as possible in order to maximize rewards in the subsequent episodes. This is analogous to the context-based framework discussed in Section 3.1.2, and, in fact, any memory-based algorithm is a context-based learner just by architecture.

**Meta-Training and Meta-Testing**

The meta-training scheme of $RL^2$ significantly differs from the general meta-RL meta-training. One meta-episode is defined as interacting with a particular MDP $M_i$ for $K$ episodes. The experiences $X^i_{\text{train}}$ update the policy parameters $\theta$ in the standard RL manner. For standard RL algorithms such as TRPO Schulman (2015) or PPO Schulman et al. (2017), this means to sample single transitions $(s, a, r, s')$ from the replay buffer $\beta_i$ containing all experience in $X^i_{\text{train}}$, and use the corresponding batch for updating $\theta$ in several epochs of gradient descent.

Throughout MDP-specific learning, the parameters $\theta$ remain unchanged. Instead, the hidden state $h_t$ of the RNN updates through the gathered experience. It is reset at the beginning of each MDP-specific training but gets preserved throughout the $K$ episodes - what is the major difference to simply modifying standard RL algorithms with an RNN agent. Conditioning the policy $\pi_\theta(\cdot|h_t)$ on that hidden state, one yields an MDP-specific adaptation scheme like the inner learning of PEARL, since the hidden state $h_t$ functions as the representation $z$ of the context $c$. An MDP-specific training-test split does not exist, as the inner learning performance is validated throughout MDP-specific training.

In contrast to its meta-training, the meta-testing scheme of $RL^2$ is analogous to the meta-RL meta-testing presented in Section 2.2. The policy parameters $\theta$ are fixed, while the hidden state adapts during $K$ episodes of MDP-specific learning. Again, this is very similar to PEARL, where only context variables $z$ adapt throughout meta-testing.

### 3.2.2 Variational Bayes-Adaptive Deep RL

In $RL^2$, the RNN has to cover all architectures theoretically needed for the optimal strategy $\pi_i^*$ of each MDP $M_i$ of the MDP distribution $p(M)$. But a single network is naturally limited in the number of strategies it can represent, what probably limits the generalization capability of the algorithm Ben-Iwhiwhu et al. (2022) and results in poor asymptotic performance Gupta et al. (2018). This aligns with the findings of Ni et al. (2022) that actor-critic RNN-based standard RL performs better when two distinct networks for actor and critic are used . At the same time, the exploration-exploitation dilemma leads to poor reward during exploration, which is nevertheless required for exploiting the gained knowledge in order to maximize future rewards. Altogether, this motivates the idea of separating the exploration process from the later exploitation, e.g., by distinct exploration and exploitation networks as presented in Stadie et al. (2018) for modifying MAML and $RL^2$, or as done in other works like Norman & Clune (2024), Liu et al. (2021), or Zhang et al. (2021).

Optimal exploration means collecting the most meaningful context possible, while exploitation requires optimally extracting the most essential information from that context in order to adjust the behavior respectively and maximize future rewards. How well the environment was explored is thus determined by the quality of the context Norman & Clune (2024), while a good representation of that context is substantial for good exploitation. This motivates leveraging context encoders as, e.g., done in gradient-based algorithms like PEARL Rakelly et al. (2019), DREAM Liu et al. (2021), MAESN Gupta et al. (2018), or MetaCure Zhang et al. (2021), to inform the policy network with meaningful context embeddings $z$. However, it may be desirable to process the whole context sequence rather than single transitions, i.e., utilizing memory architectures for such context encoders.

This subsection discusses Variational Bayes-Adaptive Deep RL (VariBAD) Zintgraf et al. (2020), the landmark RNN-based task-inference method on the development path leading to ADA. The main idea of VariBAD is to incorporate a context-based Variational Auto-Encoder (VAE) into the memory-based meta-RL frame-

work and abuse its context embeddings to represent task uncertainty and reconstruct MDP dynamics. The former is used to directly incorporate task uncertainty into the decision-making of the policy, while the latter enables the usage of model-based RL at task level. This way, VariBAD combines ideas from context-based task-inference (like presented for PEARL in Section 3.1.2) with the RNN-based meta-RL framework introduced in the previous subsection.

VariBAD outperforms $RL^2$ in both a dynamic grid world and the Mujoco benchmark Todorov et al. (2012). Exhibiting almost Bayes-optimal exploratory behavior that leads to higher returns during adaptation Zintgraf et al. (2020). While both methods can adapt within a single episode, $RL^2$ demonstrates slower and less stable learning, and other meta-learning methods like PEARL perform poorly after just one episode of adaptation Zintgraf et al. (2020). However, the Bayes-optimality of VariBAD has neither been shown nor proven across more sophisticated task distributions, and Melo (2022) even shows VariBAD to be unstable in out-of-distribution test tasks within the Mujoko environment, which indicates worse generalization than implied by the original publication. This has led to modifications of VariBAD, such as HyperX Zintgraf et al. (2021), which enhances meta-exploration by distilling policies from random networks to improve performance in distributions of sparsely rewarded MDPs, and MELTS Bing et al. (2024) specifically designed for effective zero-shot performance on non-parametric task distributions through task-clustering. Both these methods utilize modifications also used in the Adaptive Agent (i.e., distillation and task-selection), which are hence described in Section 3.4.

### VariBAD Paradigm

Besides a common policy $\pi_\theta$, the main component of VariBAD is the VAE architecture that consists of a context-encoder RNN $f_\varphi$, a decoder $g_\psi$ and a variational distribution $p_\varphi(z|c_t^i)$ as the output of the RNN encoder. The VAE decoder is an ordinary neural network $g_{\varphi_{R,T}}$ with two heads, one representing a common transition function $\mathbb{T}$, the other a common reward $R$. The common policy $\pi_\theta(\cdot|s, p_\varphi(z|c))$ is an ordinary deep neural network parameterized by $\theta$. Both the common policy and the decoder receive the output distribution $p_\varphi(z|c)$ from the VAE encoder, so that, in practice, both these components are blocks of the same architecture, where the encoder output distribution $p_\varphi$ is fed into the policy head $\pi_\theta$ and the model dynamics head $g_\psi$. This way, the VAE distribution $p_\varphi$ represents "task uncertainty" Zintgraf et al. (2020).

Analogous to PEARL, the parameters $\varphi, \theta$ and $\psi$ update on meta-level, while inner learning corresponds to the implicit change in the VAE distribution $p_\varphi(\cdot|c_t^i)$ resulting from updated context $c_T^i$. Using this task belief as an input, the common policy as well as the common dynamics $R$ and $\mathbb{T}$ implicitly adapt to the current, initially unknown MDP $M_i$. Although this setting makes off-policy RL possible, the original publication Zintgraf et al. (2020) utilizes on-policy algorithms. However, Dorfman et al. (2021) present a respective modification, that is based on leveraging offline data collected by MDP-specific agents in order to extract meta-knowledge for meta-updates of $\varphi$, $\theta$, and $\psi$.

### Meta-Training and Meta-Testing

After carefully adjusting the meta-RL and BRL paradigms of Section 2.2 to the architecture of VariBAD, adapting the meta-training and meta-testing schemes is straightforward. Analogous to PEARL's meta-training scheme, the meta-parameters $\varphi$, $\theta$, and $\psi$ do not adjust throughout the $K$ episodes of MDP-level training, while the replay buffer $\beta$ stores the resulting trajectories $\{c_t^i\}_{t=1,...,h_i}^{i=1,...,N}$ for meta-level gradient descent updates. The corresponding meta-loss $\mathcal{L}_{\text{meta}}$ consists of two major components:

1. The RL loss which simply is the accumulated loss over all trajectories, and

2. the ELBO loss, that itself consists of two components, a reconstruction loss for the decoder and a KL-divergence keeping the encoder output $p_\varphi$ close to its update. [12]

The task inference happens at the output stage of the VAE encoder, since the encoder output distribution $p_\varphi$ functions as the meta-learned prior. The corresponding posterior is updated at each timestep via variational

---

[12]For a detailed derivation of the ELBO-loss and all corresponding components, the curious reader is referred to Sajid et al. (2021).

inference. However, the corresponding scheme is rather technical and hence not shown here. Instead, the curious reader is referred to the original work in Zintgraf et al. (2020) or other works about task inference like Sajid et al. (2021).

The meta-testing of VariBAD is analogous to that of PEARL. All meta-parameters are fixed, while only the task uncertainty $p(z|c)$ is updated.

## 3.3 Transformer-based Meta-RL

The transformer network architecture was first proposed by Vaswani et al. (2017) as a language-to-language model i.e., as a model translating one language into another. As induced by its title, Vaswani et al. (2017) show that "attention is all you need" by outperforming all other state-of-the-art language-to-language algorithms by a good margin. This success motivated the developement of many transformer modifications further boosting performance, e.g., the TransformerXL Dai et al. (2019) used in ADa. Simultaneously, several researchers developed modifications of the initial transformer for tasks different from language-to-language translation, e.g., Vision Transformers for computer vision Han et al. (2023), visual Li et al. (2024a), Xiao et al. (2023) and semantic segmentation Strudel et al. (2021), forecasting Lim et al. (2021), and RL Hu et al. (2024). This way, they applied transformers to various fields like medicine Xiao et al. (2023), e.g., for RNA prediction Chaturvedi et al. (2025) fault detection e.g., in manufacturing Wu et al. (2023), bioinformatics Zhang et al. (2023a), automated driving Hu et al. (2022), and many other application fields. In RL, transformers close the gap between simulation and reality in locomotion control Lai et al. (2023), act beneficially in various IoT environments like Smart-Homes or industrial buildings Rjoub et al. (2024) and learn from a vast amount of (sub)-optimal demonstrations Liu & Abbeel (2023), Lee et al. (2023), Reed et al. (2022). Since each of the various extensions and modifications of the vanilla transformer builds up on the attention mechanism proposed in the original publication Vaswani et al. (2017), the original transformer architecture is broadly presented and discussed in Appendix B, while this section focuses on the utilization of transformer architectures for meta-RL.

### 3.3.1 Transformers are Meta-Learners

Instead of translating a sentence from one language into another, a context-based RL agent (like PEARL or VariBAD) "translates" context trajectories of the form (14) or their context embeddings into the current action $a_t$. Such a "translation" does not require a decoder as the current action can directly be derived from a policy head. But it is likely that the single transitions of a context trajectory have strong dependencies to each other or the time they occurred, so that it is additionally desirable to contextualize them respectively. This motivates to use the transformer architecture, e.g., in simple, gradient-based meta-learning algorithms like MAML (Section 3.1.1). Hence, different works present modifications for detecting faults in bearings Li et al. (2024b), improving short-term load forecasting in scenarios where different clients require federated learning Feng et al. (2025), and forecasting stock prices Chen et al. (2025).

However, transformers additionally have a demonstrable long-term memory of up to 1500 steps into the past Ni et al. (2023), what particularly motivates to also utilize transformers in memory-based meta-learning Melo (2022), Grigsby et al. (2023), Xu et al. (2024), Shala et al. (2024b), Shala et al. (2024a). TRMRL Melo (2022) is such a transformer architecture tailored for meta-RL. It is the last landmark transformer-based meta-RL algorithm on the development path to the Adaptive Agent, and fulfills all necessary properties of a meta-learner Melo (2022). The main idea of TRMRL is to replace the RNN in the $RL^2$ paradigm with a transformer agent that is additionally, initialized in a certain way to improve initial training stability.

Although PEARL demonstrates slightly better sample efficiency in locomotion tasks, TRMRL outperforms MAML, PEARL, RL2 with PPO as an inner learner, and VariBAD in the Mujoco environment Todorov et al. (2012) wrt. sample-efficiency, adaptation speed and particularly OOD performance. This superior performance is even more significant in the more sophisticated Meta-World environment Yu et al. (2020). The only exceptions are Mujoco tasks with high task uncertainty, where VariBAD slightly outperforms TRMRL due to its explicit uncertainty modelling. This motivates combining a transformer-based meta-RL agent with Bayesian inference like in PSBL Xu et al. (2024), or ADA. the former outperforms TRMRL as well as other baselines such as MAML, $RL^2$, PEARL or VariBAD in terms of asymptotic and OOD performance

on sparse and dense reward tasks - an observation that becomes even clearer when the distribution shift increases Xu et al. (2024). The latter explicitly builds up on the variBAD paradigm and achieves human-like performance. However, before presenting the Adaptive Agent in the subsequent section, the following paragraphs discuss the TRMRL paradigm.

**TRMRL Paradigm**

TRMRL extends $RL^2$ by a transformer architecture i.e., by self-attention, while otherwise completely mimicking the $RL^2$ paradigm. This means that the model weights are also only updated on meta-level, while inner learning corresponds to the model activations triggered by the collected experience. With respect to these two stages, Melo (2022) show that

1. the fast adaptation of the multi-head attention serves as a task representation mechanism, since each self-attention head contextualizes the embeddings of the context $c_t^i$ collected in the current MDP $M_i$ up until the current time step $t$.

2. the meta-learned model weights function as a long-term memory by design, through which the respective MDP can be identified and acted upon accordingly.

They hypothesize that any MDP can be represented by a distribution over working memories:

$$z_t = \sum_{t=1}^{T} \alpha_t f_\theta(o_t) \tag{18}$$

where $o_t$ is a single transition of the form $(s_t, a_t, r_{t+1}, s_{t+1})$, $N$ is the trajectory length, $f_\theta$ is a learnable, arbitrary linear function, and $\alpha_t$ are coefficients summing up to 1. In fact, the context scores of the last transformer encoder sub-block naturally have the form of such a task representation when given the context trajectory $c_t^i$ of the form (14) as input at a particular time point $t$: Reformulating the self-attention softmax term (20) for a particular transition $o_t$ and the multiplication with the corresponding value vector $v_t$, one yields representations for the coefficients $\alpha$ and the linear function $f$ in (18) - the former through rewriting the softmax term, the latter by a sum over entries of the value vector. For the rather technical full derivation of that MDP representation property, the motivated reader is referred to the original publication Melo (2022). For the scope of this work, it is sufficient to conclude that this MDP representation property creates the implicit objective to "learn to make a distinction among the tasks in the embedding space" Melo (2022), i.e., the implicit goal to learn how to identify tasks through the collected experience. In fact, Melo (2022) additionally prove that any transformer self-attention layer minimizes the task uncertainty based on the current context $c_t^i$, i.e., that every transformer sub-block finds the best representation of the gathered information. In each transformer sub-block, the previous representation of $c_t^i$ is recombined through the dense layers before the next self-attention layer persists in the form of a task representation. This way, the representation of $c_t^i$ gets episodically refined. This process "resembles a memory reinstatement operation" Melo (2022), i.e., an operation to reactivate long-term memory in order to identify the current task and act accordingly.

**Meta-Training and Meta-Testing**

In RL, transformers are often unstable during training Hu et al. (2024), especially in the beginning Melo (2022). And while more sophisticated algorithms like AMAGO Grigsby et al. (2023) stabilize training by incorporating off-policy learning into their transformer architecture, TRMRL addresses this problem right at the beginning of meta-training by utilizing T-Fixup initialization Huang et al. (2020). This initialization scheme is especially designed to avoid learning rate variation and layer normalization during early training, and the ablation studies of Melo (2022) particularly show its usefulness. Besides that, the meta-training and meta-testing schemes of TRMRL directly derive from $RL^2$.

## 3.4  The Adaptive Agent

Transformers still struggle with "credit assignment" Ni et al. (2023), i.e., with assigning current actions to rewards later on in the planning horizon - a problem that even increases when the data complexity is high Ni et al. (2023). This particularly motivates to utilize model-based RL, where the agent can base its decisions on the predictions of the dynamics model. The only landmark algorithm on the development path towards ADA utilizing model-based RL for inner learning is VariBad. It additionally models task uncertainty into its paradigm and, hence, Melo (2022) already identify the possibilities of combining the VariBad paradigm with a transformer architecture. The Adaptive Agent is this transformer-based modification of VariBad. It combines a large transformer architecture with self-supervised learning techniques in order to develop a generalist agent applicable to a vast number of "downstream tasks".

ADA builds on other, similar approaches aiming for a generalist agent, i.e., RL foundation model, like GATO Reed et al. (2022) or the work of Team et al. (2021). It achieves human-like few- and zero-shot performance (in terms of generalization and adaptation speed) for single and multi-agent tasks on an even more complex and dynamic extension of the XLAND environment Team et al. (2021), a "vast open-ended task space with sparse rewards" for single and multi-agent tasks. In multi-agent downstream tasks, the different Adaptive Agents clearly share labor and actively cooperate to improve their reward, while in single-agent tasks it is able to use one single episode of expert demonstrations in order to significantly improve performance Team et al. (2023). This makes ADA superior to the work of Team et al. (2021) and other generalist agents like GATO Reed et al. (2022) which exclusively learn from demonstrations. However, ADA is not generally superior to human-level performance, as there are single- and multi-agent downstream tasks neither humans nor ADA are able to solve.

ADA's transformer memory length, i.e., the length of the context $c$ used as model input, as well as its model and task pool sizes, scale demonstrably well, particularly when being scaled together with each other Team et al. (2023). For example, increasing the task pool, i.e., sampling a larger number of distinct tasks from a task distribution with more complex tasks, always has a positive effect on the agent's median performance, especially in the few-shot setting, but this effect is even bigger when model size or memory also increase. The same holds for the model size and the transformer memory length: larger models as well as models with longer context windows obtain better median performance in any $K$-shot setting, particularly when the task pool also increases. On a log-log plot, the effect of increasing the model or context window size additionally scales roughly linearly with the number $K$ of shots. But only if the task pool is sufficiently large. Otherwise the model overfits the small task pool. All these findings additionally hold true in the 20th quantile i.e., in the 20,This indicates a more stable training and highlights why Bommasani et al. (2021) identify scaling as the key factor of what makes foundation models so powerful.

Although ADA is capable of handling varying context lengths, it, like most RL transformers, assumes the dimension of observations to be constant throughout tasks. However, as ADA particularly utilizes an additional observation encoder prior to its transformer architecture, it suffices to exchange only that encoder when observation spaces change. This motivates AMAGO Grigsby et al. (2023), a more flexible approach that particularly utilizes an observation encoder mapping each context to a fixed-sized representation, so that the encoder is the only required architectural change across experiments. This way, AMAGO is particularly designed to incorporate off-policy actor-critic RL in combination with dynamic, long-term contexts, so that it can train agents from multiple rollouts in parallel from more distinct, long-term contexts, which also makes it applicable for multi-task RL.

Since approaches like AMAGO, GATO, and ADA clearly are foundation models, they are computationally expensive during meta-training Shala et al. (2024b), which motivates hierarchical transformer architectures like HTRMRL Shala et al. (2024a) or ECET Shala et al. (2024b). They use additional self-attention sub-blocks to process cross-episode data in order to further increase sample efficiency and reduce model complexity. In contrast to the intra-episode self-attention sub-blocks utilized in TRMRL, AMAGO, and ADA, these cross-episode blocks set whole episodes of experience in context to each other, so that the context mechanism of self-attention is also leveraged at the episode level. Among other model reduction techniques like distillation, this is a promising direction for future work in the field of foundation models and generalist agents.

**The ADA Paradigm**

Analogous to the VariBad paradigm presented in Section 3.2.2, the ADA paradigm consists of a policy $\pi_\theta$, a decoder $g_\psi$, and a context encoder $f_\varphi$ whose output is the variational distribution $p_\varphi(z|c_t^i)$ representing task uncertainty. However, the ADA paradigm modifies these components:

- Prior to the transformer architecture, a ResNet architecture preprocesses the visual 3D observations received from the XLand environment.

- The memory-based context-encoder is a transformer architecture. [13] More precisely, the TransformerXL-architecture Dai et al. (2019), a demonstrably more efficient and powerful modification of the vanilla transformer whose architecture allows it to model long-range dependencies.

- The architecture following the transformerXL mimics the Muesli algorithm Hessel et al. (2021), a computationally efficient, demonstrably powerful RL approach that combines model-based and model-free RL. This requires a critic network estimating the value (11) of the current state as well as a dynamics model for planning. The latter predicts policy and critic network outputs along with future rewards and transitions.

In addition to these modifications, the number $K$ of adaptation episodes varies (i.e. $K = \{1, 2, \ldots, 6\}$). It is, hence, an additional input of the context encoder.

**Meta-Training and Meta-Testing**

In contrast to TRMRL, the training scheme of ADA does not consist of a particular initialization scheme. Instead, layer normalization and gating techniques stabilize the transformer during training. Besides that, the VariBad meta-training scheme is extended by two self-supervised learning techniques:

1. An automatic curriculum (ACL) selecting tasks on the frontier of the current agent's capabilities for efficient learning, and

2. distillation for kickstarting the training process.

The subsequent subsections describe both these techniques in more detail.

### 3.4.1 Automated Curriculum Learning:

Autocurriculum Learning (ACL) leverages the idea from human learning that training tasks should not be too hard or too easy for the current state of learning. The main idea originates from curriculum learning (CL), Dansereau (1978), where curricula are hand-crafted to guide the learning process of a standard learner (e.g., a neural network) on a single task. Such curricula were successfully applied in several supervised learning tasks like NLP, computer vision, medicine, NAS, and RL - see e.g., Wang et al. (2022b), Soviany et al. (2022), Gupta et al. (2021b), for a detailed overview. The main idea of ACL is to automatically prioritize (sub) tasks at the frontier of the agent's capabilities during training, instead of manually selecting them. This ensures that the agent is consistently challenged and can progressively improve its skills Portelas et al. (2020), so that sample efficiency and generalization performance can be significantly improved Dennis et al. (2021), Jiang et al. (2021), Jabri et al. (2019), Wang et al. (2022b).

There are several ACL methods (see e.g., Portelas et al. (2020) for a detailed overview). One of the earliest approaches is Teacher-Student learning Matiisen et al. (2020), where a second model (the teacher) is simply used to select subtasks for the standard learner (the student) during training. For meta-RL, there also exist various techniques, e.g.,

- Simple, but quite static methods like domain randomization (e.g., Volpi et al. (2021)), where environments are generated randomly, but independently of the current policy's capabilities.

---

[13]Note that theoretically any transformer architecture can be used that is suitable for RL.

- More flexible approaches like min-max-adversarial (e.g., Adv), where environments are created adversarially, i.e., to challenge the agent as much as possible.

- Sophisticated paradigms like min-max-regret, e.g., Protagonist Antagonist Induced Regret Environment Design Dennis et al. (2021), where an adversary aims to maximize the regret between the protagonist (the agent) and its antagonist (an oponnent. The latter is assumed to be optimal.

For min-max-regret, the regret is defined as the difference between the value of the protagonist's and the antagonist's action. This way, the adversary selects the simplest tasks the protagonist is not yet able to solve Dennis et al. (2021). This makes min-max-regret superior to min-max-adversarial, which, due to its objective, tends to generate tasks too difficult for the agent or even unsolvable Dennis et al. (2021). However, ADA leverages and compares two other ACL methods during meta-training:

1. NOOB-Filtering Team et al. (2021) maintains a control policy that takes no actions. After rolling out ADA and the control policy for ten episodes in a new task, it is selected for meta-training if: ADA's performance is neither too good nor too bad, the variance among trials is sufficiently high between episodes to indicate proper learning, and The control policy performs poorly enough to indicate the relevance of proper decision making.

2. Robust Prioritized Level Replay (PLR) Jiang et al. (2021) maintains a "level buffer" of tasks with a high learning potential. It samples tasks from that buffer with a probability of $p$, while, otherwise, sampling a new task from the task distribution $p(T)$.

   The regret of a task estimates its learning potential. A task is added to the level buffer, if its regret is higher than those of the other level buffer tasks. This constantly refines the level buffer. In ADA, several fitness metrics like policy and critic losses approximate the task regret.

In comparison to meta-training with uniformly sampled tasks, both no-op filtering and PLR, improve ADA's sample efficiency and generalization as well as its few- and zero-shot performance Team et al. (2023). Additionally, PLR slightly outperforms NOOB filtering, especially when the number $K$ of adaptation episodes is higher. However, any other approach from the ones described above might work just as well.

### 3.4.2 Distillation

The notion of distillation was first proposed by Hinton et al. (2015). It generally refers to model compression techniques transferring knowledge from a large, pre-trained teacher model $T$ [14] to a smaller, more efficient student model $S$ Rusu et al. (2016). Hence, distillation can be referred to as a transfer-learning approach Mansourian et al. (2025). Classically, the student is trained to learn from the teacher's demonstrations, i.e., it learns to predict the teacher's softmax output distribution (in the discrete case) that is smoothed by choosing a sufficiently high softmax temperature parameter in order to yield soft targets Rusu et al. (2016), Mansourian et al. (2025), Hinton et al. (2015). In RL, one of the best-known distillation approaches is policy distillation Rusu et al. (2016) (also known as immitation learning), where the student policy is trained to imitate the teacher's action distribution. Other widely used approaches include value function distillation and dynamic reward-guided distillation (see Xu et al. (2025) for further details).

Generally, the two main questions in selecting a knowledge distillation scheme are what knowledge to transfer and which architectures to respectively choose for teacher and student models Mansourian et al. (2025), Gou et al. (2021). ADA utilizes dynamic reward-guided distillation, where the student policy is guided by the teacher policy rather than particularly mimicking it. It contains an additional distillation loss throughout the first four billion meta-updates that consists of the KL-divergence between student and teacher policy action distributions, as well as a $L^2$-regularization term like in Schmitt et al. (2018). This way, the student policy can also explore states different from those visited by the corresponding teacher model, which was trained from scratch with a model size of 23 million parameters.

---

[14] It is also possible to use multiple teachers for different subtasks Schmitt et al. (2018). However, these teachers can be seen as one teacher model consisting of different task-specific components.

For the sake of comparison, Team et al. (2023) train four different models: a large (256 million parameters) and a small (23 million parameters) model both, with and without, distillation. They show the larger distilled model to strongly outperform the smaller one as well as the large ordinary model, while the smaller ordinary model outperforms the larger one. This not only indicates a significant positive effect of distillation on model performance but also highlights the necessity of distillation for largely scaled (foundation) models.

## 4 Discussion

Meta-learning describes the paradigm of learning how to learn. But, as reality poses a vast amount of potential tasks, the question necessarily arises, which general knowledge to acquire. The earlier algorithms presented and discussed in the previous sections primarily focused on meta-learning the skill to make decisions in RL environments by pre-training model weights (MAML), basing the decisions on the currently explored context (PEARL) or the current context-based belief over which task they are acting in (VariBAD) with manually designed architecture modifications. Transformer-based agents are meta-learners just by design. Although they were not specifically designed to be meta-learners, these models learn how to learn just by emergence.

The path towards general intelligence often looks like that Bommasani et al. (2021): While earlier approaches are hand-designed for particular purposes, later ones yield more general, unsupervised, and unguided intelligence, learning different skills emergently. The consequence is a shift towards "homogeneity", i.e., the usage of one model architecture for several different tasks or task distributions Bommasani et al. (2021). In other words, as soon as a model architecture generalizes better, it is used for more general purposes. The path towards the Adaptive Agent presented in Section 3 examplifies this developement. Even within the transformer-based development path, one can clearly observe this shift towards homogeneity: While early transformer-based meta-RL architectures like TRMRL require hand-designed architectures for different observation spaces, generalist agents like ADA and AMAGO abstract this problem by observation space-specific encoder blocks. Observing this scheme of developments (like e.g., outlined by Bommasani et al. (2021) and Clune (2020)), one gets an idea of how future developments might look like: As, from a theoretical point of view, any kind of skill can be meta-learned, meta-learning does not have to focus particularly on the task-specific adaptation itself. The easiest example is meta-learning hyperparameters, - an approach that occurred quite early in the timeline of meta-learning developments with algorithms like $\alpha$-MAML Behl et al. (2019) or PEARL Rakelly et al. (2019). But even far more complicated skills like neural architecture search or designing curricula or populations of evolutionary algorithms can be meta-learned. The last of the following subsections discusses such approaches and sets them in context to the path towards general intelligence.

### Relevance of Meta-Learning

With the shift from manually designed, relatively simple to understand meta-learning algorithms like MAML or PEARL to largely scaled, blackbox foundation models like GATO Reed et al. (2022), ADA Team et al. (2023), or AMAGO Grigsby et al. (2023), the question arises whether the paradigm of meta-learning is actually still of relevance. Largely scaled foundation models from companies like DeepMind, OpenAI, or Meta can be applied to a vast amount of in- and out-of-distribution downstream tasks of different types - as e.g., done for the LAMA model in Rentschler & Roberts (2025) for RL tasks - without even thinking about data quality, distribution shifts, or the meta-training paradigm. However, such foundation models require a vast amount of computational power, even for simple downstream rollouts without model updates. During training, they necessitate scaling of their model size, the task pool, and the task complexity Team et al. (2023), Bommasani et al. (2021). Such an amount of data and computation power is not available for all researchers, companies and people - even when techniques like distillation and ACL decrease the model size and boost the training progress. Moreover, in recent years there has been a significant shift from publicly available models with revealed source code trained on open-source data towards secretly training models of unknown sizes and shapes on largely collected datasets that are hidden from the public - and, as such, hidden from public control and revision. But how models behave highly depends on the data they were trained on, so that this privatization of model training comes with a high risk of unexpected, harmful behavior

Bommasani et al. (2021), along with the societal risk of an increasing gap between institutions with a high amount of computational power and data and others who cannot utilize large resources.

Modern research can, regardless of the available computation power, focus on different niches of application Togelius & Yannakakis (2024) as well as on developing more efficient architectures like hierarchy transformers Shala et al. (2024a), Shala et al. (2024b), or on gaining a better understanding of blackbox models, their behaviour, the reasons for their failures, and the societal and ethical impact their application has Togelius & Yannakakis (2024). The latter can support the development of even better general problem solvers Clune (2020), so that collaboration between university researchers and Big Tech companies is probably beneficial for both Togelius & Yannakakis (2024), Bommasani et al. (2021).

**The Need for Specialized Meta-Learners**

In application, agents must often be deployed on small edge devices with a very limited amount of computational power that does not allow for generalist agents but might benefit from meta-trained models that adapt fast. Such problems likely cover only a very narrow, low-dimensional task distribution that does not require generalist agents. In other words, largely scaled transformer-based architectures often go far beyond what is needed to solve a particular problem. This is probably why most meta-learning applications are still based on simple, sample-efficient gradient-based algorithms like MAML, although they have worse OOD performance (see e.g., the applications for medicine Tian (2024), Alsaleh et al. (2024), Tian et al. (2024), Ranaweera & Pathirana (2024), Naren et al. (2021), biomass energy production Zhang et al. (2025), or fault diagnosis Lin et al. (2023)). However, understanding the advantages and drawbacks of the various components of the landmark algorithms presented in the previous section can support applied researchers in manually designing the best fitting meta-learner for their particular problem. For example, autonomous driving requires extremely fast adaptation, very good zero-shot performance and high robustness and reliability, while sample efficiency does not really matter, especially during training. In such an application, gradient-based algorithms are most likely the wrong choice, while memory-based Bayesian inference learners like VariBAD or ADA are much more promising. In contrast, medical applications typically have a very limited amount of data available per patient, so that they particularly require high sample efficiency. For such applications, gradient-based meta-learners are a good first choice, as they are simple and more sample-efficient. This especially holds true for off-policy meta-RL algorithms like PEARL. To assist the encouraged reader with the choice of a meta-learning algorithm for application, Table 3 provides an overview of the main advantages and drawbacks of the developments discussed along the timeline of the previous section.

**Comparability**

There is a lack of comparability among the landmark algorithms presented in the timeline of the previous section. The utilized performance measures differ between the different works and are often not clearly defined, an issue this work aims to address by defining general performance measures in Section 2.1. In addition, different works utilize different benchmarks with different properties and of different complexity. Thereby, the development of those benchmarks follows the development of meta-RL algorithms itself: Early landmarks like MAML, RL$^2$, or PEARL were mainly tested on simple grid-world environments, the Atari benchmark Bellemare et al. (2013), or the Mujoco engine Todorov et al. (2012). But Fakoor et al. (2020) identified that those benchmarks are too simple to evaluate actual adaptation, which motivates more generalistic, complex, and sparsely rewarded environments like Meta-World Yu et al. (2020), Crafter Hafner (2021) or XLand Team et al. (2021). This lack of homogeneity makes it difficult for researchers and developers to compare different algorithms and modifications. However, this is a general problem of ongoing research and development and can hence, only be tackled by several empirical studies.

**The three Pillars of the Path towards General Intelligence**

The development path towards human-like or even superior generally intelligent agents consists of three pillars Clune (2020):

- Meta-learning algorithms that can be viewed as the "evolution of intelligence" itself,

Table 3: Advantages and disadvantages of the different components introduced along with the landmarks in Section 3. The last column lists algorithms containing the respective component.

| Component | Advantages | Drawbacks | Algos |
|---|---|---|---|
| Gradient-based | • Fast adaptation, easy to implement | • Weak generalization and OOD performance | MAML, FO-MAML |
| Context-based | • Can dynamically adjust to tasks | • Depends on the quality of context | CAVIA, DREAM, MAESN, Meta-Cure PEARL |
| Bayesian Inference | • Better uncertainty quantification, adaptive | • Complex implementation | MAESN, Meta-Cure, PEARL, VariBAD, MELTS |
| RNN-based | • Sequence processing, context retention | • Low Sample-efficiency | $RL^2$, VariBAD |
| Transformer-based | • Powerful contextualization | • Large amount of data and computation power required | TRMRL, ADA |
| ACL | • Improves training efficiency | • May be costly to implement | MELTS, ADA |
| Distillation | • Reduces model size while maintaining performance | • Potential loss of precision | HyperX, ADA |
| Model-based RL on Task-Level | • Utilizes prior experiences for quick adaptations | • Higher model complexity | VARIBAD, ADA |
| Generalist Agents | • Can handle a wide variety of tasks | • High computation cost even for rollouts | Various RL algorithms |

- meta-NAS Pereira (2024), Hospedales et al. (2022) creating the "physical body" of the gained "intelligence," and

- the generation of more meaningful, complex, and challenging environments that function as the "world" the "physical body" of the learned "intelligence" is acting in.

While research in the meta-learning community mainly focused on the pilar of learning algorithms and, secondly, on meta-NAS approacheS, meta-environment research is very sparse Clune (2020). However, as the growing complexities and capabilities of recent developments come with the necessity of a further increase in complexity for benchmark environments, the attention most recently shifts towards this research field by combining meta-RL with self-supervised learning techniques. This is reflected in the ADA paradigm, where initial learning is guided by a teacher model (Distillation) and an automatic curriculum (ACL) selecting tasks on the frontiers of the agent's capabilities. The latter is, however, a skill that can be meta-learned, so that a (teacher) model learns to create a personalized curriculum for any student model Portelas et al. (2021), Xu et al. (2023). The corresponding Meta-ACL paradigm is inspired by classroom scenarios, where a teacher teaches several students at once, ideally with respect to their personal capabilities and special needs.

Additionally, meta-design of environments can be inspired by examining the evolution on Earth Clune (2020). Instead of developing algorithms generating open-ended environments like XLand Team et al. (2021) and meaningful rewards, one can aim for agents that explore on their own, i.e., out of curiosity Alet et al. (2020) or through intrinsic reward (see e.g., Zhang et al. (2021)) in environments providing a high variety of different niches to adapt to. The former corresponds to techniques like novelty search Lehman & Stanley (2011), while algorithms like quality diversity Pugh et al. (2016) can be utilized to generate high-variety environments.

Considering the current work of Big Tech companies, a very likely future path of meta-learning and foundation model development will be towards combining these pillars respectively. The most recent landmark of that kind is the Evolution Transformer Lange et al. (2024) of DeepMind. It combines the pillars of meta-learning and meta-NAS by learning how to design the population of RL agents (individual) for evolutionary RL algorithms. In other words, on top of the evolutionary approach, the Evolution Transformer learns how the population of agents (breed of creatures) must be designed to have the highest chance of survival as a whole, i.e., it focusses on the continued existence of the species rather than on maximization the performance of single individuals.

Combining the different pillars results, along with a potential increase in model capability, in several potential ethical, societal, and economic concerns that are very difficult to predict. When models are "taught to play god" - even in a very limited world like the currently existing environments - it is essential to track their behavior, the basis on which they make their decisions and the harm they can potentially cause from different views like law Chan et al. (2024), ethics, economy and various others. In this respect, the future development might be examined and analyzed analogously to that of foundation models in Bommasani et al. (2021), where a large group of various researchers of several domains and backgrounds collaborate in order to identify potential risks, challenges and benefits of the current shift towards largely scaled foundation models.

## 5 Conclusion

In this comprehensive survey, we provided the timeline of meta-RL developments from foundational algorithms like MAML and RL$^2$ to ADA, a largely scaled generalist agent. Through detailed mathematical formalizations of meta-learning and meta-RL paradigms, we addressed the lack of detailed formalism within the literature and laid the groundwork for understanding the paradigms and training schemes of the landmark algorithms presented along that timeline. Based on that formalized knowledge, we were able to highlight the paradigm shift towards transformer architectures and largely scaled foundation models through the usage of self-supervised learning techniques like distillation and automated curriculum learning. Looking ahead, we examined the three pillars of general intelligence and identified a trend towards the automated generation of environments through meta-learning of self-supervision and evolutionary approaches. Besides offering valuable insights, this highlighted the constantly growing need for collaborative, interdisciplinary teams of researchers that permanently analyze the behavior of generalist agents and their ethical implications for our society.

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

# A  Terminology

In the literature, meta-learning is often confused with the notions of transfer learning (TL) and multi-task learning (MTL). Although these concepts share quite some similarities - there are even algorithms exhibiting overlapping characteristics among them Upadhyay (2023) - the confusion mainly arises from a lack of clear definition. Therefore, the following paragraphs broadly introduce TL and MTL focusing on a clearly defined paradigm. For a more detailed consideration, the reader is, nevertheless, referred to other works like Upadhyay (2023) particularly focusing on meta-learning, MTL, TL and their overlaps.

## A.1  Transfer Learning

The significance of the concept of Transfer Learning notably heightened following its integration with deep neural networks. Landmark advancements in computer vision, exemplified by architectures such as AlexNet Krizhevsky et al. (2012), GoogleNet Szegedy et al. (2015), and EfficientNet Tan & Le (2019), are pre-trained on the extensive ImageNet dataset before employing them to tasks such as image classification, segmentation, and object detection. These models demonstrated substantial improvements over traditional machine learning algorithms in terms of adaptability, sample efficiency, and few-shot performance. Motivated by these achievements, transfer learning has also been effectively applied in various other domains, including speech recognition (e.g. Wav2Vec Schneider et al. (2019)), healthcare applications Esteva et al. (2017), and reinforcement learning (RL). The latter itself distributes into various applications as knowledge transfer is

used to close the gap between simulation and reality in robotics Yu et al. (2019), learn from demonstrations to incorperate expert knowledge Ravichandar et al. (2020), Sosa-Ceron et al. (2022), or adapt to new rules and gaming mechanics in game play using reward shaping OpenAI et al. (2019). All these various sub fields of TL and transfer RL present distinct challenges and are hence areas of ongoing research. Interested readers are thus encouraged to explore specific surveys on TL Niu et al. (2020), Zhuang et al. (2021), Tan et al. (2018), Panigrahi et al. (2021), and transfer RL Zhu et al. (2023), Zhao et al. (2020) for deeper insights.

**Transfer Learning Paradigm**

The main idea of TL is to extensively pre-train a model $F_\theta$ on a source task $T_1 := \{\mathcal{L}_1, \mathcal{X}_1, \mu, \mathbb{T}_1, h\}$ and transfer the acquired knowledge by fine-tuning $f_\theta$ on a related target task $T_2 := \{\mathcal{L}_2, \mathcal{X}_2, \mu, \mathbb{T}_2, h\}$. Mirroring human learning, this approach leverages knowledge from pre-training to enhance few-shot performance and learning speed on the target task $T_2$. For example, one can extensively practice inline skating in the summer to prepare for skiing when it snows for only a few days during winter. Therefore, the source task $T_1$ typically contains abundant, well-labeled data $\mathcal{X}_1$, while the target task's data $\mathcal{X}_2$ often is sparse, of low quality or expensive to obtain. Additionally, the goal encoded by $\mathcal{L}_1$ can differ from that represented by the loss function $\mathcal{L}_1$ or there may be a change in domain dynamics $\mathbb{T}$ that needs to be learned quickly. The latter is of particular importance in RL problems, where changes in environment dynamics can cause severe issues; for example when an agent is supposed to drive a car and the terrain becomes significantly more slippery. A change in the loss function can, however, become problematic if the loss $\mathcal{L}_2$ is considerably more computationally expensive to calculate as this slows down learning.

One can also define the TL paradigm with the possibility of several source tasks. However, this is rather a technical difference as one can define $T_1 = \bigcup_{i=1}^{N} T_1^i$, where $T_1^i$ are the different source tasks. If, however, more than one target task exists, one typically uses the notion "downstream tasks". This is the case in more recent developements, where TL, along with a vast amount of data and self-supervised learning approaches, led to much more powerful and broadly applicable models, namely foundation models.

**Foundation Models**

Foundation models serve as a common basis from which many task-specific models are built through adaptation Bommasani et al. (2021), i.e. they are pre-trained by a vast amount of source tasks as described above, before being fine-tuned to different downstream tasks. As a consequence, foundation models are inherently incomplete: they rather serve as a "foundation" for fine-tuning to various applications than as static general problem solvers. This way, the adaptation of foundation models follows the meta-testing paradigm described in Section 2.1, while their training is that of a TL algorithm not following the meta-training paradigm of Section 2.1. As they are meant to be trained on a distribution $p(T)$ of training tasks rather than one single source task before being fine-tuned to a (potentially different) distribution of downstream tasks $p_{\text{test}}(T)$, they can be viewed as the "meta-level version" of TL.

TL is the key technique that makes foundation models possible, but scale is what makes them so powerful Bommasani et al. (2021). This fact probably arises from the universal approximator property of neural networks: larger models can represent more complex functions of higher dimension, although they also tend to overfit more easily. The latter is the reason why besides model size, the data (e.g., the task pool in meta-learning) is also required to scale Team et al. (2023), Bommasani et al. (2021). However, as the amount of data increases, the data quality can not neccessarily be maintained. Instead, self-supervised learning techniques Gui et al. (2024), Ericsson et al. (2022) are used to learn patterns in available, but potentially unlabeled, data, so that foundation models are often defined as "large-scale transfer-learning models pre-trained via self-supervised learning" Bommasani et al. (2021).

There was a paradigm shift towards foundation models that get fine-tuned for downstream tasks within the recent years Bommasani et al. (2021). For example, almost all state-of-the-art NLP models are now adapted from one of a few foundation models, e.g., Bert Devlin et al. (2019), GPT3 Brown et al. (2020), GPT4 OpenAI et al. (2024), LLAMA Touvron et al. (2023). Similarly, foundation models were developed for various applications like agriculture Yin et al. (2025), medicine Zhang & Metaxas (2024), Liang et al. (2025),

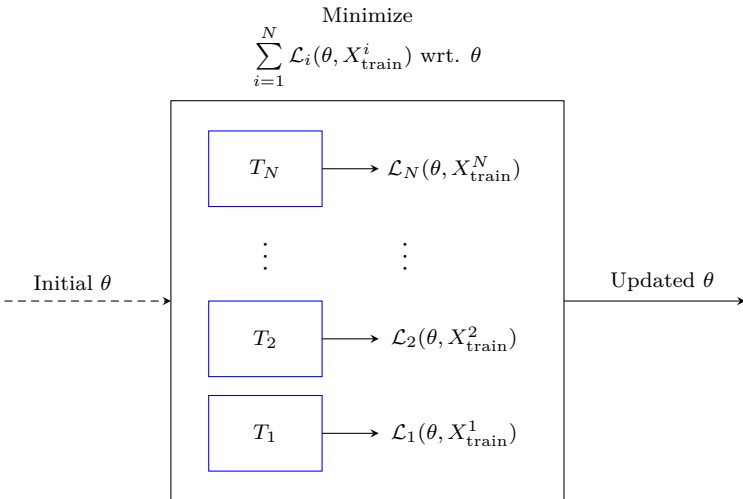

Figure 8: General Multi-Task-Learning Paradigm:

Moor et al. (2023), or robotics Firoozi et al. (2025). In RL, foundation models have been used for reward engineering Ye et al. (2024), Wang et al. (2024b), e.g., to generate rewards from only text description and the visual observations yielded from the environment Wang et al. (2024b). Additionally, Rentschler & Roberts (2025) successfully fine-tune the LLAMA Touvron et al. (2023) foundation model on RL downstream tasks without RL-specific training with remarkable results. As a consequence, several works combine NLP and RL paradigms into RL from Human Feedback (RLHF), which is itself an ongoing and vast research field. We refer interested readers to particular RLHF surveys like Kaufmann et al. (2023), Casper et al. (2023), Chaudhari et al. (2025).

### A.2 Multi-Task Learning

The concept of Multi-Task Learning (MTL) was first introduced by MTL as the formalization of the idea of training models on multiple tasks simultaneously. The main idea is to train one single model among several similar tasks simultaneously in order to find valuable, general feature representations that can be leveraged to improve task-specific performances.

Applying the paradigm of MTL to different neural network architectures like Transformers Torbarina et al. (2024) or Bayesian networks Lazaric & Ghavamzadeh (2010), ..., led to remarkable successes in neural language processing Chen et al. (2024), neural architecture search Gao et al. (2020), Pasunuru & Bansal (2019), segmentation Bischke et al. (2019), Zhou et al. (2021), Tang et al. (2023), medicine Bi et al. (2008), Zhang et al. (2023b), Zhou et al. (2021), and robotics Arcari et al. (2023), Gupta et al. (2021a), Deisenroth et al. (2014), to list only a few. We, however, refer interested readers to MTL-focused surveys like Zhang & Yang (2022), Zhang & Yang (2018), Yu et al. (2024) or Multi-Task-RL (MTRL) specific works like Vithayathil Varghese & Mahmoud (2020) or D'Eramo et al. (2024) for a broader overview of MTL and MTRL algorithms. This section rather focuses on MTL and MTRL paradigms to help readers distinguish them from TL and meta-learning or transfer-RL and meta-RL respectively.

### Multi-Task Learning Paradigm

Formally, in MTL a model is jointly trained to solve a given number $N$ of tasks $(T_1, T_2, \ldots, T_n)$ sharing some common structure. The corresponding training process mimics standard learning on task-specific level, but with parameters being shared throughout the tasks. A meta-level does not exist and hence there is no two-staged training process. Instead, the MTL paradigm aims to solve all $N$ tasks equally well. Figure 8

visualizes the resulting optimization problem. It can be formalized as

$$\text{Minimize} \quad \sum_{i=1}^{N} \mathcal{L}_i(\theta_i, X_{\text{train}}^i) \quad \text{wrt. } \theta \tag{19}$$

where $\theta_i \in \mathbb{R}^{d_i}$ denotes the task-specific parameters reserved for each individual task.

The task specific parts of the model typically are task-specific heads with one or more layers, so that the model parameters $\theta$ consist of task-specific parameters $\theta_i$ for each task $T_i$ as well as common knowledge denoted by $\varphi$, but all in one single model. The performance of the MTL model is evaluated on task-level only, i.e. on the task-specific test sets $X_{\text{test}}^i$. A meta-test on unseen tasks does not take place.

In contrast to the meta-learning paradigm described in Section 2.1, the MTL paradigm does not treat the parameters $\theta_i$ of each task as individual parameter sets. They are rather considered as the different parts of one bigger set of parameters $\theta = \{\theta_0, \theta_1, \ldots, \theta_N\}$ with $\theta_0 \in \mathbb{R}^{d_0}$ denoting the common knowledge over all tasks. A deep neural network can, for example, learn some feature representations shared between all different tasks to extract the relevant information of its data before fine-tuning to the respective task. Such a NN theoretically consists of different blocks: A general part encoding all the information shared between tasks, and a smaller head for each individual task.

**Multi Task Reinforcement Learning**

In Multi-Task Reinforcement Learning (MTRL) one develops a policy for several similar but different tasks at once. Analogous to the general MTL paradigm, the common knowledge is about how to process the information the current state provides, which action has which effect on the environment and what are general dynamics shared among tasks. At the same time, rewards and transitions differ between the different MDPs, what requires for task-specific heads to decide for an action out of the (commonly) processed state information. Since the notion of a policy is rather a theoretical concept, this means the same neural network can function as the agent in the different MDPs with a task-specific head for each MDP. A second part of potentially common knowledge is task identification. However, compared to the more general meta-RL paradigm, this is rather simple: The model always faces the same $N$ tasks. Hence, most algorithms simply use one-hot encoding.

In the literature, MTRL is often confused with Multi-Agent learning - where multiple agents act within the same environment while trying to increase individual or collective rewards - or Multi-Actor Learning - where multiple agents with the exact same goal collect experiences and share them for faster learning. But these two scenarios correspond to the single-task setting. The former extends the notion of single-task to multiple agents, while the latter only uses multiple instances of the environment in order to parallelize training.

### A.3   Comparison

For differentiating between the three different paradigms of TL, MTL and Meta-Learning, it is important to understand that they are answers of increasing levels of abstraction Upadhyay (2023) to the question of how to exploit knowledge from previously learned tasks to solve new ones in only a few shots. TL, as the solution with the lowest level of abstraction, simply transfers the knowledge gained in one task (the source task) to a second one (the target task), while Meta-Learning, as the solution with the highest level of abstraction, generally tries to extract the core information to learn any task of a distribution $p$ in a few shots. In other words, Meta-Learning also extracts information from the source task(s) to use them as a prior for the target task(s), but this information can be much more general, and it results from the outer optimization rather than from solving the source task(s) in particular - a difference becoming even more unclear when foundation models get pre-trained on several source tasks in order to be meta-tested afterwards.

The most confusion in the literature is between MTL and meta-learning Upadhyay (2023) since both aim to solve several tasks as well as possible rather than exclusively focusing on one single target task (like in TL). While meta-learning iteratively and sequentially learns a number $N$ of tasks not necessarily fixed and, afterwards, tests the thus acquired knowledge on test tasks unseen during training, MTL trains a

fixed number $N$ of tasks simultaneously and without testing the resulting model parameters $\theta$ on unseen tasks. The trained MTL model is, instead, evaluated on the validation set $X_{\text{test}}^i$ of each of the learnt tasks, $T_1, \ldots, T_N$ which corresponds to the standard machine learning test phase but for $N$ tasks at the same time. In this way, MTL rather focuses on best solving the learnt tasks than on adapting to new ones as fast and good as possible.

MTL can consist of tasks of different families, such as classification, regression, or segmentation, while meta-learning is normally reduced to one family of tasks represented by the distribution $p$ over tasks of the form (2). And, most importantly, MTL does not consist of an outer learner extracting prior knowledge (or meta representations). In this specific aspect, MTL is more similar to transfer learning, where prior knowledge between tasks is only transported implicitly via the model parameters. However, the TL paradigm particularly requires a target task to transfer the gained knowledge to, while the MTL paradigm does not specifically include such an option. This further highlights the similarity between TL and meta-learning as well as the difference between meta-learning and MTL.

# B    Attention is all you need

The following paragraphs broadly discuss the architecture of the vanilla transformer Vaswani et al. (2017). The goal is to give the reader an intuition how self-attention "creates context" and "focusses" on important information on the example of language-to-language translation. For a detailed description of the vanilla transformer architecture the interested reader is nevertheless referred to the original paper or Alammar (2018).

The vanilla transformer consists of an encoder and a decoder, which both divide into six 2- or 3-layer sub-blocks respectively [15]. Metaphorically speaking, the encoder translates an input sentence like

<p style="text-align:center">The child plays football because it likes it very much.</p>

into "machine language", i.e. word embeddings, while the decoder translates these embeddings back into the desired output language. However, a word-by-word transformation from the input language into "'machine language "' and back into the output language preserves no information about what word the first "it" refers to and to which word the second "it" is related to. One needs a possibility to preserve the context between the words and, hence, the semantic structure of the sentence while encoding it. Similarly, while decoding, the syntactic structure of the sentence must be transformed so that it fits the grammar of the output language. For example, a German sentence has a totally different grammatical structure than its English counterpart with the same semantic meaning. In their original publication, Vaswani et al. (2017) solved these problems using two techniques: Self-attention layers and positional encodings. The following paragraph discusses the former, while the latter is broadly discussed further below.

## B.1    Self-Attention

In a self-attention layer, every word embedding $x_i$ $(i = 1, 2, \ldots, n)$ of a $n$-word input sentence $x$ like the one above gets a query, key and value projection matrix $Q_i, K_i$ and $V_i$ assigned to it. These matrices are parameters of the self-attention and, as such, get learned during training. They project a word embedding $x_i$ into query, key and value vectors $q_i, k_i$ and $v_i$ by multiplication. The "context" between words is then considered in the following way:

- Every word embedding $x_i$ gets "compared" to every other word embedding $x_j$ by multiplying every query vector $q_i$ with every key vector $k_j$ in a dot product. One can interpret the result as the context between the corresponding words of the input sentence since, mathematically, the dot product puts the query and key vectors in geometrical relationship to each other.

- For each word embedding $x_i$, the result of its query-key multiplication with every other word embedding (including itself) is a vector representing the attention scores of the word embedding $x_i$ to

---

[15]This number of six sub-blocks for encoder and decoder is rather an architectural choice by Vaswani et al. (2017) than a necessity of the vanilla transformer.

all other word embeddings $x_j$. This vector of attention scores is fed into a softmax function to yield the respective attention weights summing up to one.

- At last, these "attention weights" are multiplied by the value vector $v_i$, resulting in a weighted sum over attention weights with weights $v_i$ that directly depend on the learned projection $V_i$. Hence, the model can learn which attention scores $q_i \cdot k_j$ are more or less important by adjusting the values of the value projection matrix $V_i$ respectively.

In practice, one typically simultaneously calculates all query-key pairs of the embedded input sentence $x$:

$$\text{Attention}(x) := \text{softmax}(\frac{QK^T}{\sqrt{d}}) \, V, \tag{20}$$

with the hyperparameter $d$ as the dimension of key, value and query vectors, $\sqrt{d}$ as the scaling factor and $K$, $V$ and $Q$ being the respective key, value and query matrices. The scaling factor $\sqrt{d}$ keeps the dot product logits small so that the soft-max does not slip into saturation. This maintains a proper gradient flow and thus stabilizes training.

Each encoder sub-block of the vanilla transformer consists of a self-attention layer so that the context between words (and hence the semantic structure of the sentence) is preserved throughout the whole embedding process. Encoding the above example sentence, the model can learn through the query-key multiplication $q_1 \cdot k_2$, that the first word "The" refers to the next word "child". At the same time, this connection is almost meaningless to the semantic of the whole sentence so that the model possibly learns to assign a very small value to the value vector $v_1$.

The self-attention layers of the decoder sub-blocks look slightly different to those described above. Since decoding the embeddings into output language is a sequential manner, no word can be set into context to words following later on in the sentence. Therefore, the corresponding query-key multiplications $q_i \cdot k_{i+a}$ with $a = 1, \ldots, n - i$ are set to minus infinity by construction so that the respective softmax results in zero attention weights.

In their original work, Vaswani et al. (2017) implement multi-head-attention with the number of heads as a hyperparameter and show this to further boost performance. Multi-Head Attention duplicates the word embeddings $x_i$, projects them into the smaller subspace of each head, [16] executes the Self-Attention of the form (20), , concatenates the outputs of all heads to one single output matrix, and projects them back into the original output dimension $d$ by another linear layer transformation.

### B.2 Positional Encoding

In the self-attention (20) the order of input tokens does not matter and hence the order of the words of the input sentence is not automatically preserved throughout encoding. However, the order of words in a sentence is quite important, especially as syntactic structure of input and output languages might differ, e.g. when translating English into German. This is why, Vaswani et al. (2017) include a positional embedding for each word embedding $x_i$ in the vanilla transformer architecture that has the same length as the embedding itself. The form of the positional encoding $p_i$ is a hyperparameter of the vanilla transformer. In their original work, Vaswani et al. (2017) combine different sin and cosine functions, but state that they have tried different positional encodings without a major loss in performance. Other transformer variants even learn this positional embedding function along with all the above model parameters.

---

[16] the queries, keys, values of each head are smaller-dimensional vectors.

