# OpenReview forum: "Meta-Learning and Meta-Reinforcement Learning - Tracing the Path towards Deep Mind's Adaptive Agent"
_TMLR — Rejected by TMLR_

### Review · Reviewer_pJcN · 2025-08-21

**Summary Of Contributions:**

1. The topic of meta-RL is important, and a survey paper can be highly beneficial for researchers entering this area.

2. However, the current version lacks coherence across sections, and the relative importance of different parts is not clearly conveyed.

**Audience:**

Yes

**Audience Explanation:**

Meta-RL is an active and growing research area, and a dedicated survey could provide useful insights and orientation for both newcomers and experienced researchers.

**Claims And Evidence:**

Yes

**Claims Explanation:**

1. Although the title suggests a focus on meta-RL, the discussion on this topic does not begin until page 24. The earlier sections are primarily about meta-learning in general, which feels disconnected from the paper's stated purpose. For example, Section 4, which reviews the timeline of meta-learning, appears somewhat redundant in this context.
2. The paper does not sufficiently explain its unique contribution compared to existing survey papers. In particular, it should explicitly address why meta-RL warrants a dedicated survey distinct from the broader meta-learning literature.
3. While some representative algorithms are described, their connections to the overall landscape of meta-RL are not clearly established. It would be helpful to provide a unifying framework that shows how these algorithms fit together.
4. Section 3 discusses transfer learning and multi-task learning in contrast to meta-learning. While this material is informative, it seems more relevant to a survey on meta-learning in general rather than a paper focused specifically on meta-RL.

**Requested Changes:**

1. Clearly state the novelty of this work compared to existing surveys on meta-learning.
2. Bring the focus on meta-RL earlier in the paper. For example, the discussion of meta-learning could be shortened, as it has already been well covered in other surveys.
3. Correct a typo on page 4: "… and $h \in \mathbb{N}$".
4. Correct the caption of Figure 8, which contains an extra ":" at the end.

---

### Review · Reviewer_RzLZ · 2025-09-06

**Summary Of Contributions:**

The claimed contributions of the paper are to: 1) provide a detailed formulation of meta-learning and meta-RL, 2) discuss the most common performance measures in meta-learning, 3) distinguish meta-learning from multi-task learning and transfer learning, 4) provide a detailed history of meta-learning, and 5) discuss open problems and future directions.

**Audience:**

No

**Audience Explanation:**

I do not believe that this paper presents any findings. I also do not feel like this paper makes a clear argument for why its claimed contributions are even needed. There are already a number of highly cited survey papers on the topic of meta-learning.

I found the writing quality to be quite poor throughout. There is no clear narrative that the paper pushes for and I did not feel that there was any useful synthesis of information. The high-level categorizations provided are very obvious to those familiar with the field. I also do not feel like it will be useful in its submitted form for new researchers looking to learn about meta-learning because it is just way way too much scattered information. It does not break things down in a digestible way for readers. In fact, most paragraphs meander around a number of topics. Obviously, doing this effectively would be quite difficult, but I feel the related work sections of some of the seminal papers in meta-learning provide a more concise and easy to understand description of the key ideas in the field.

**Broader Impact Concerns:**

I think the broader impact has been adequately considered within this paper.

**Claims And Evidence:**

No

**Claims Explanation:**

Going through the contributions listed above, I will now provide my thoughts on each one:

1) This paper provides definitions of each topic as they have already been defined in the literature. There is no attempt to provide clarity or synthesize information. There is not a single definition of meta-learning provided, but rather each definition that has been proposed within the meta-training and meta-testing paradigm is gone through. I feel this paper does not make a contribution to the literature regarding the formulation of meta-learning and ignores the formulations of meta-learning that tackle the more natural setting of continual learning without data that has been pre-segmented into tasks for different phases of learning. See [1] and Section 5.3.2 of [2].

2) This is not a contribution to the literature either. The seminal papers on this topic all clearly define the performance metrics they consider and there is not a new synthesis or conclusion regarding performance metrics that are orthogonal or complimentary etc.

3) This has been done in the related work section of nearly every paper on this topic.

4) I was extremely disappointed by this part. The title of the paper is literally "The Timeline of Meta-Reinforcement Learning - From the
beginnings to the Adaptive Agent". The authors do at least cite Schmidhuber 1987, but then go on to basically say that it didn't gain traction until 2017 and actually fully ignores the first 30 years of a topic that has existed in the field of AI for 38 years. I can only imagine how insulted by this framing Schmidhuber would be. Literally all of his contributions to this area are ignored. He has many interesting papers on this topic in the 90s alone that in my opinion are vital to the history of meta-learning. There are also a lot of important contributions by Turing Award winners such as Yoshua Bengio and Richard Sutton that are ignored from this period. What about Samy Bengio's 1993 PhD thesis on meta-learning in neural networks? I do certainly agree that interest in AI overall has been higher in recent years and the number of papers and citations is large in this period, but feel that this paper falls woefully short of its title. I think that if the authors actually dive deeper into the literature from this 30 year period, they will find that research of this time embraced a far more ambitious form of meta-learning than the recent work on meta-training and meta-testing. In my view, this is part of the reason for the seemingly declining relevance of the kind of meta-learning considered by the authors that is discussed at the end.

5) Again, I don't believe there were any new insights presented here about open problems or future directions.

[1] "A History of Meta-gradient: Gradient Methods for Meta-learning" Richard Sutton, 2022.

[2] "Towards Continual Reinforcement Learning: A Review and Perspectives" Khimya Khetarpal, Matthew Riemer, Irina Rish, and Doina Precup, 2022.

**Requested Changes:**

I believe that this paper needs to be entirely rewritten to contribute to the TMLR community. For a history of meta-learning the exclusion of the majority of that history from the survey is unacceptable. There are many papers from this period that should be discussed in detail, particularly considering the amount of attention this paper pays to more recent papers of far less significance to the field. If this is not done, the title of the paper must be changed. However, I am frankly not sure what contributions to the community it would be left with. Considering the scope of the paper, it is far too long and should aim to champion a more concise and clear narrative that will relieve readers of the burden of learning about too many different papers and ideas in order to grasp the key ideas in this field.

The writing is consistently of poor quality throughout and the paper is too long for me to provide detailed feedback. Instead, let me just focus on the number of issues within the contributions list itself to be illustrative of what I mean. Line 1: I believe the authors mean "manyfold" rather than "manifold" from geometry. The authors after the colon on line 1 have a hanging "It." This is either an unprofessional formatting mistake or an attempt to distribute the word "It" at the beginning of every contribution. I guess the latter makes the most sense given the mismatch between lack of capitalization and punctuation for each bullet, but this formatting is just so unusual that it distracts readers and lacks any clear purpose. First bullet: aren't "clean" and "detailed" contradictory statements? Second bullet: "It well defines" is a very awkward read. Third bullet: "closeliest"? Fourth bullet: "the earliest landmarks" are MAML and RL^2? Final bullet: it should be " connects them" to resolve the pronoun agreement issue.

---

> ### Author Response · Authors · 2025-09-10
>
> Thank you for your feedback. We understand that our work is too long to be reviewed in detail, and we will shorten it accordingly in our next version. As this paper aims to help researchers entering the field dive more deeply into it, the chosen level of detail might not suit that of an advanced researcher in meta-learning or related fields. Quite the contrary, this survey focuses on certain concepts and landmark algorithms to highlight the path toward a generalist agent like ADA.
> We understand the reviewer’s concern regarding the large number of citations. However, from our perspective, this is a core function of a survey paper: to provide a high-level overview of the field—especially for readers less familiar with it—while referencing key works and surrounding developments that allow interested readers to explore specific sub-topics in more depth.
> We are open to adjusting the presentation if certain sections feel overloaded or insufficiently focused.
>
> We would like to clarify that it was not our intention to provide a complete account of the entire history of meta-learning or meta-RL. Indeed, as you also pointed out, the manuscript is already quite extensive.
> Our goal instead was to consolidate the essential knowledge required to understand ADA within a single paper.
> Striking the right balance between completeness, digestibility, and effectiveness is admittedly challenging, and we very much appreciate your recognition of this difficulty.
> Regarding your points:
> 1. "This paper provides definitions of each topic as they have already been defined in the literature."
> Yes, that was exactly what we promised in the introduction: to "present a clean and detailed mathematical formalization of meta-learning and meta-RL."
> However, from our perspective, this is much more valuable than a narrative definition of "learning how to learn," as it provides a more detailed description of the meta-learning notion than a simple intuition.
>
> Moreover, we acknowledge the reviewer’s important point regarding alternative formulations of meta-learning that align with the continual learning perspective, where tasks are not pre-segmented but arise naturally over time. This setting indeed represents a highly relevant extension of the classical meta-training/meta-testing paradigm.
> faced with the complexity underlying algorithms like ADA, we, however, want to rather break things down into an understandable paradigm that develops throughout the journey from MAML to ADA.
> Although these concepts, notions, and formulas are (in part) covered by other works, none of them are complete for our scope.
>
> 2. We would like to clarify that our intention was only to formalize metrics that are often only briefly mentioned in figures or  captions in the literature. In doing so, we aimed to provide the community with a clear, unified mathematical framework that captures what is frequently taken for granted but rarely written down in a systematic way.
> 4. We acknowledge that our introduction and title may have unintentionally suggested that we intended to provide a comprehensive history of meta-learning as a whole. This was not our goal, and we will revise both the title and contribution statement to more accurately reflect the actual scope of the paper. We agree that influential earlier contributions, of course deserve explicit acknowledgment, as by the two pages historical review of Richard Sutton you kindly provided.
> They simply do not have a direct influence to understanding the Adaptive Agent we chose as our main motive in this survey.
>
> 5. We conclude that a careful understanding of meta-learning and meta-RL paradigms is still highly relevant—perhaps more than ever—to understand what is learned emergently. This is one more motivation for detailing all the formulas in section 2, especially those for performance measures, which might be utilized to assess the emergent capability to meta-learn of foundation models/generalist agents.
> The large amount of other frequently cited meta-learning surveys, as you stated, underscores the ungoingly high importance of meta-learning, in contrast to a "seemingly declining relevance of the kind of meta-learning," you claimed.
>
> Additionally, we respectfully disagree with the statement that no new insights were provided in the discussion of open problems and future directions. Most other surveys about meta-learning or meta-RL are three or more years old (including the ones you provided).
> Consequentially, none of them discusses ADA, any other generalist agent, the emergence property or the three pilars of general AI.
>
> To adress the above concerns, we will:
> 1. Change the title to "Meta-Learning and Meta-Reinforcement Learning - Tracing the Path towards Deep Mind's Adaptive Agent."
> 2. Sharpen our formulations in the introduction.
> 3. Condense section 2 to a more reasonable amount.
> 4. Move section 3 to the appendix.
> 5. Revise section 4 in alignment with the feedback provided in the review above.

---

### Review · Reviewer_UQ6R · 2025-09-06

**Summary Of Contributions:**

The paper's main contributions are (1) a timeline of the main contribution to meta-learning reinforcement learning, and (2) a framing of meta-learning reinforcement learning in the more general universe of meta-learning.

The paper's main weakness is the attempt to do too much, which leads to actually doing too little; that is, too little synthesis, insights, and crisp, intuitive comparing-and-contrasting of the very long list of papers and their main results. All of this is due to excessive scope, poor organization, and poor writing. In its current form, the paper has a low density of insights; what is more, these insights are sprinkled all over the place, which puts the onus on the reader to make the effort in collecting and connecting them (which should the main goal of the authors).

**Additional Comments:**

Please be consistent in the spelling of meta-learning, meta-learning RL, etc

**Audience:**

Yes

**Audience Explanation:**

Both meta-learning and meta-learning reinforcement learning are active fields of research with potentially impactful applications.

**Claims And Evidence:**

Yes

**Claims Explanation:**

The authors' claims are accurate in the sense that the paper presents a timeline of the main contribution to meta-learning reinforcement learning, and frames meta-learning reinforcement learning in the more general universe of meta-learning. As this paper is a review of past work, it does not make novel claims that would need theoretical and/or empirical evidence.

**Requested Changes:**

This paper requires significant work to improve both its organization and writing.

Organization-wise:
A. Streamline the introduction: in its current version, there are 6 paragraphs before you get to the paper's contributions. Please shrink these 6 paragraphs to at most 3, so that the readers don't have to through two pages of generalities before getting a glimpse about what the paper os about
B. Streamline Section 2: in its current form, it uses 14 pages to provide very little value. The reader should expected to be familiar with the basics of Machine Leaning (ML) and Reinforcement Learning (RL), which will allow you to avoid explaining concepts like training/validation/test sets. Figure 1 is too trivial to include, while figure 2 could be easily replaced with a 3-sentence paragraph that intuitively explains Meta-Learning. In this reviewer's opinion, this section should be light on notation and heavy on intuition and illustrative examples. It is hard to justify why a reader should be meandering between trivial concepts and quite a bit of notation just to get to section 2.2, where the Meta-RL is finally presented.
3. Section 3  could be easily in an APPENDIX, or, alternatively, be simplified to half a page early in Section 2. It is simply unacceptable that you only reach "the meat of your paper" at page 21 (i.e., Section 4)
D. Section 4:
D.1 Figure 9, 12, and 13 do present the promised timelines. However, they should be supplemented by one table each, in which you would compare-and-contrast all the key properties of the algorithms in the corresponding figure. Any algorithm that does not deserve a full line in the table should be removed from the figure, too, especially if it is barely even mentioned in the text. For example, the sentence before 4.1.2 provides the motivation behind Taming-MAML and DICE, but it does not discuss them; in this reviewer's opinion, if the details are not worth discussing, nor is the motivation.
D.2 Overall, the three tables to be added (per above) will represent the core of the paper. Rather than discussing the "bolded algorithms" (in the three figures) independently of each other, "the meat of the paper" should be an intuitive, systematic compare-and-contrast discussion of the differences between the various algorithms
D.3 A major miss of the paper is a lack of information wrt empirical evaluations. What tasks is each algorithm better suited for? Why? Again, compare & contrast.
E. the discussion on the relevance of meta-learning should not be "hidden" on page 37, but rather on the first page. This subsection should also be streamlined: what is your conclusion on this topic?

Writing:
F. Please remove the passive voice: the paper uses a large amount of passive voice, which are, by definition, ambiguous
G. Please spell/grammar check the paper, as it includes many spelling and grammar errors
H. often times, the authors use very long, intricate sentences (2+ lines long) that are ambiguous and hard to follow. Please do an extensive cleanup and make sure that the sentences are no excessively long or intricate; just a few illustrative examples:
- please remove the last phrase in the abstract ("the connection of meta-learning to the path towards general intelligence"): is this a paper aboutMeta-RL? or meta-learning & AGI? Please stay on the message and do not over-promise
- page 1:  Introducing "Model-agnostic Meta-Learning" (MAML), Finn et al. (2017a) founded the class of gradient-based meta-learners by abstracting the mere concept of gradient-based learning to a meta-level, where model parameters are aimed to be placed a priori in such a way that task-specific gradient-based learning has the most impact (see section 4.1)
- page 2: Since the topics related to or associated with meta-learning are widely spread and often difficult to distinguish from meta-learning itself or each other, several works surveying meta-learning and related topics have been published that categorize different sub-classes of meta-learning Beck et al. (2023b), Vettoruzzo et al. (2024) or differentiate between the numerous related topics Vettoruzzo et al. (2024), Barcina-Blanco et al. (2024), Upadhyay (2023).
- page 5: the first sentences in the bottom paragraph
- page 15 - last sentence
- page 20: first sentences in last paragraph
- page 21: the paragraph before 4.1
- page 23: last paragraph
- page 36:  Classically, the student is trained to learn from the teacher’s demonstrations, i.e. it learns to predict the teacher’s softmax output distribution (in the discrete case) that is smoothened by choosing a sufficiently high softmax temperature parameter in order to yield soft targets Rusu et al. (2016), Mansourian et al. (2025), Hinton et al. (2015).

---

> ### Author Response · Authors · 2025-09-09
>
> Thank you very much for this valuable and detailed feedback. We are aware of the large size of the manuscript and will use your suggestions to improve readability, structure, and quality in the revision phase. In accordance with your points:
>
> Introduction: In alignment with the other reviews, we will shorten and sharpen the introduction in order to make our scope more clear:
> 1. We derive meta-learning from standard learning and then transfer the notions and formulas to meta-RL.
> 2. We use this paradigm as a consistent lens through which we present the landmarks on the path to ADA. This is meant to mirror section 2. We want the paradigm to be the main motive, the line to follow, when new concepts arise alongside each new landmark.
> 3. At the end of that timeline, we examine ADA as one of the most powerful generalist agents. Since we have clarified other concepts by then, we can focus on the boosting techniques like distillation and ACL.
> 4. These techniques, together with other current developments, form the discussion in section 5.
> The main idea is to build a bridge between detailed formalization and intuition by applying the paradigm to landmarks.
>
> Writing style: We fully acknowledge the issues with long sentences, grammar, and passive voice. We will perform a thorough language revision to make the paper more concise and accessible.
>
> Section 2: we understand the concern about length and redundancy, although we believe that a careful derivation of training/validation/testing (and their meta-level counterparts) is useful. No other work considers this rather practical detail. If mentioned, meta-validation and meta-testing are often not distinguished from each other.
> When we first dealt with applying meta-learning, we had to give a lot of thought to this, even though standard learning was clear to us.
> However, we will significantly condense this section by shortening the introductions to standard learning and standard RL as you suggested.
>
> Section 3: We initially felt this section was important for completeness, but since all reviewers suggested otherwise, we will move it to the appendix to keep the main paper focused on the contributions.
>
> Timeline section: We will supplement the figures with comparison tables that summarize and contrast the key properties of the algorithms. Algorithms that are not discussed in detail in the text will be removed to increase the dense of insights and improve readibility.
> However, please be aware of the fact that the question of when to use which meta-RL algorithm cannot be generally answered (please see our discussion about that in section 5).
> We will solve this problem by presenting algorithm strengths and weknesses in the tables you suggested.
>
> We are confident that these changes will address the reviewer’s concerns about scope, synthesis, and organization, and will result in a much clearer and more impactful paper.

---

### Comment · Action_Editor_yXiS · 2025-08-26
**Review extension until 5 Sept.**

Hi all,

given that the amount of material that reviewers need to cover for this submission is very large, even by TMLR standards (39 pages of main paper content), I have informally granted a review extension until end of next week, that is, Fri 5 Sept. (11:59pm AoE).

I want to thank the authors for their patience, and the reviewers for their work and engagerment (and thanks to pJcN for delivering their review already).

Best,
  AE

---

### Author Response · Authors · 2025-09-18
**Major Revision**

We, again, thank the reviewers for their valuable feedback.
We have fully revised our work in terms of language, structure and contents.
This reduced the main paper content by almost ten pages,

---

> ### Comment · Action_Editor_yXiS · 2025-09-20
> **[Reviewer action required]**
>
> Dear reviewers, please read the authors' responses and revised manuscript and reply to the authors' rebuttal. We are aiming to enter the final review phase (final reviewer discussion and final recommendation) by 3 Oct. the latest.
>
> If you have no further questions or comments, and you have acknowledged the authors' response, you can enter your final recommendation via OpenReview. Please have a look at [TMLR's acceptance criteria](https://jmlr.org/tmlr/acceptance-criteria.html), which focus primarily on scientific correctness and audience fit, and not on novelty or potential impact of the work (of course, these criteria need to be applied in the context of a review-style paper).
>
> Thank you all for your effort and engagement, and thanks to the authors for their responses and preparing a major revision of the manuscript.
>   AE

---

### Decision · Action_Editor_yXiS · 2025-10-06

**Recommendation:** Reject

**Audience:**

Yes

**Audience Explanation:**

In principle I think large parts of TMLR's audience would be interested in the topic of this paper. At its current execution and focus experts may criticise the degree of depth and rigor, whereas newcomers to the field may find themselves overloaded with the complexity and volume of the material. I would still rate the answer to the question as a 'Yes', but I think this could easily be raised to a strong and enthusiastic 'Yes'.

**Claims And Evidence:**

No

**Claims Explanation:**

The paper aims to provide a comprehensive overview and review of developments in meta-RL over the last decade(s), as well as relevant works from the area of meta-learning, culminating in DeepMind's Adaptive Agent (culminating in a chronological sense; the paper provides much broader and much more background than what is needed to understand the Adaptive Agent paper). This review-style endeavour is an arduous and demanding task (reflected in the paper's length of 29 main paper pages and 51 pages in total). A well-executed paper achieving this task is a valuable and interesting contribution to the community and TMLR in particular. Unfortunately, the current manuscript does not quite meet the bar, as criticised by the reviewers (1 Reject, 1 Leaning Reject, 1 Leaning Accept; all after the rebuttal). I agree with some of the reviewers' criticism (hence the 'No' in the answer field above), and I do appreciate the authors' response and updated manuscript. I enjoyed reading parts of the manuscript, and overall I think a lot of tedious groundwork has been laid. Overall, I also believe, however, that spending more time on the manuscript to sharpen the focus (or spending a significant amount of time to keep the breadth of the current manuscript and improve the clarity and depth) is beneficial in this case and will lead to a strong and impactful paper. My recommendation therefore is to reject the paper at this stage and perform a major revision (either resubmitting to TMLR, or another venue). I do want to encourage the authors to spend more work (and sharpen the focus), as I believe that promising parts are in place already.

My personal recommendation is to either lean towards a very accessible introductory paper aimed at newcomers to the field (which means trimming the main paper a lot and focusing on clarity and simplicity, and a few key references/works), or to focus on a "more narrow" path to the Adaptive Agent aimed at experts (with full detail along this narrow path, but compressing other parts into a shorter discussion / move to the appendix). Having said that, I would also be excited about a well executed review paper that covers meta-RL with large breadth and at expert level, but the bar for doing this well and adding value to the community, and the sheer amount of work required, is very high. Think about a particular target audience, and focus on what that audience would benefit most of.


For visibility, I am paraphrasing the final comments (positive and negative) by reviewers:
One reviewer thought that the paper was significantly improved during the rebuttal. One reviewer thought that the paper aims at an important gap in the literature. One reviewer was not convinced of addressing the initial criticism with a shift in focus. One reviewer thought that the writing in many parts of the paper is still poor and thinks an entire rewrite is necessary. One reviewer thinks that the paper suffers from the tension that it would mostly benefit readers new to the area, while it seems to be written mainly for people with prior knowledge. One reviewer thinks that a paper with these aims could be valuable for the community but according to their assessment the current manuscript does not meet the bar in terms of quality.

Finally, I thank the reviewers for their time, effort, and dedication for reviewing a paper with 2-3 times the length of a standard TMLR paper.

**Resubmission Of Major Revision:**

The authors may consider submitting a major revision at a later time.